# Learning General Representations Across Graph Combinatorial Optimization Problems

## Abstract

Combinatorial optimization (CO) problems are classical and crucial in many fields, with many NP-complete (NPC) examples being reducible to one another, revealing an underlying connection between them. Existing methods, however, primarily focus on task-specific models trained on individual datasets, limiting the quality of learned representations and the transferability to other CO problems. Given the reducibility among these problems, a natural idea is to abstract a higher-level representation that captures the essence shared across different problems, enabling knowledge transfer and mutual enhancement. In this paper, we propose a novel paradigm **CORAL** that treats each CO problem type as a distinct modality and unifies them by transforming all instances into representations of the fundamental Boolean satisfiability (SAT) problem. Our approach aims to capture the underlying commonalities across multiple problem types via cross-modal contrastive learning with supervision, thereby enhancing representation learning. Extensive experiments on seven graph decision problems (GDPs) demonstrate the effectiveness of CORAL, showing that our approach significantly improves the quality and generalizability of the learned representations. Furthermore, we showcase the utility of the pre-trained unified SAT representations on related tasks, including satisfying assignment prediction and unsat core variable prediction, highlighting the potential of CORAL as a unified pre-training paradigm for CO problems.

## 1 Introduction

Combinatorial optimization (CO) is a pivotal area of study in both theoretical computer science and a wide range of applied fields, owing to its broad applicability in solving complex real-world problems, including logistics (Sbihi & Eglese, 2010), network design (Vesselinova et al., 2020), scheduling (Hwang & Cheng, 2001), and finance (Pekeč & Rothkopf, 2003). CO problems are inherently challenging due to their discrete and non-convex nature, which often leads to NP-hard complexity (Karp, 2010), with many instances requiring worst-case exponential time to solve. In response to these challenges, machine learning (ML) approaches have recently emerged in the CO domain (Bengio et al., 2021; Gasse et al., 2022), offering the potential to reduce solving times by exploiting common patterns and structures in CO instances.

Most existing ML-based approaches for CO primarily emphasize improving problem-specific representation learning to enhance task performance. While these methods can achieve high accuracy on particular tasks, the representations they learn are typically tailored to specific instances, making them non-transferable across different datasets or problem domains. Consequently, individual models must be trained for each dataset and task, limiting the potential for broader generalization and scalability. This fragmentation hinders the development of more generalizable CO models that can efficiently solve a wide range of problem types using a unified framework, as in the vision or language field (Khan et al., 2022; Min et al., 2023). Moreover, the inherent connections among many CO problems offer a compelling opportunity for unification. Since numerous NPC problems can be reduced to one another, they share a common underlying structure that can potentially be exploited for more efficient representations. The connections suggest that instead of learning problem-specific representations, a higher-level, abstract representation could be developed to capture the essence shared across different CO problems. Such a unified representation would not only enable knowledge transfer between problem domains but also facilitate mutual enhancement, as insights gained from one problem could benefit the solution of others.

In this paper, we aim to develop the general and high-level representations across CO problems to facilitate various tasks, with a particular focus on graph decision problems (GDPs), which encapsulate the core challenges of CO. Notably, from the 21 NP-complete problems identified by Karp (2010), 10 are GDPs, highlighting their fundamental importance. To achieve our objective, it is significant to effectively incorporate and synthesize features from multiple problem types. Therefore, we leverage contrastive learning, a technique widely employed for modality alignment in vision-language pretrained models (Du et al., 2022). However, applying contrastive learning to CO problems presents a significant challenge due to the inherent differences between CO problems, rendering direct application impractical. In response, we propose a sophisticated and unified training paradigm, **CORAL**, that enables effective contrastive learning across graph CO problems. Specifically, to align with the multi-modal training perspective, we conceptualize each GDP type as a distinct problem modality. To bridge the gaps among GDP types, we introduce the Boolean satisfiability (SAT) problem as a unified intermediary modality. The SAT modality is used to construct strong correspondence with other GDP types through instance transformation, thereby establishing connections among GDPs. In the training phase, instances from each GDP type are concurrently contrasted with the corresponding SAT instances, thereby fusing features across problem modalities. The contrastive-based training enables each model to learn high-level representations from multiple problem types, serving as a pre-training phase. The trained models are finally fine-tuned on specific datasets and tasks.

Extensive experiments are conducted to evaluate the effectiveness of the CORAL paradigm. First, we assess the performance of the models on standard tasks adopted during the pre-training phase, including GDP solving and satisfiability prediction, to demonstrate the superiority of the representations learned through CORAL. Subsequently, we evaluate the generalizability of the models by testing them on larger-scale instances, where experimental results indicate that the models trained using the CORAL paradigm exhibit significantly enhanced generalization capabilities. Additionally, to further highlight the practical applications of CORAL, we examine the performance of the pre-trained SAT models on related SAT-based tasks across both seen and unseen datasets during the pre-training phase. **The main contributions of the paper are as follows.**

1) We propose CORAL, a novel training paradigm designed to learn high-level representations across multiple CO problems. To the best of our knowledge, it is the first framework to leverage unified representations across different problem types.

2) We introduce SAT as an intermediate, unified modality to bridge diverse GDPs, enabling the effective learning of shared characteristics and information transfer across different problem types.

3) We conduct extensive experiments on various problems and examine the efficacy of the pre-trained representations on new tasks and datasets, illustrating the potential of CORAL as a robust and unified pre-training paradigm.

## 2 RELATED WORK

**Graph Learning for CO.** The application of machine learning to graph-based CO problems has a rich history, with recent research demonstrating substantial advancements in this domain (Khalil et al., 2017; Bengio et al., 2021; Mazyavkina et al., 2021). Most ML-based approaches for CO follow a two-stage framework: *(1) Graph representation learning*, where graph instances are embedded into low-dimensional vector spaces (Hamilton et al., 2017b; Cai et al., 2018; Chen et al., 2020a); and *(2) The utilization of these learned representations to solve CO problems* (Joshi et al., 2019; Prates et al., 2019; Sato et al., 2019). Our CORAL paradigm focuses on enhancing the first stage by proposing a more general training approach. While previous work has largely focused on designing network architectures (Kipf & Welling, 2016; Hamilton et al., 2017a; Veličković et al., 2017), our approach emphasizes the development of a training paradigm that leverages information from multiple problem types. By incorporating a contrastive learning-based strategy, CORAL aims to learn high-level, transferable representations that can be effectively applied across various CO problems, promoting a more unified and generalizable framework for graph-based CO tasks.

**Graph Contrastive Learning.** Current graph contrastive learning frameworks primarily rely on graph augmentations, which can be broadly categorized into two types: *(1) structural perturbations*, such as node dropping, edge sampling, and graph diffusion (Duan et al., 2022; Huang et al., 2023); and *(2) feature perturbations*, such as adding noise to node features (Hassani & Khasahmadi, 2020). These augmentation strategies have demonstrated effectiveness across a range of tasks, from

graph-level representations (Hassani & Khasahmadi, 2020; You et al., 2020) to node-level representations (Wan et al., 2021; Tong et al., 2021). Our CORAL paradigm moves beyond traditional graph augmentations by contrasting graph instances across multiple problem types. Instead of solely relying on structural and feature perturbations, CORAL leverages the inherent characteristics of different CO problems, enabling the model to capture higher-level characteristics.

**Solving SAT with ML Approaches.** ML-based SAT solvers can be broadly classified into two categories (Holden et al., 2021; Guo et al., 2023; Li et al., 2023): *standalone neural solvers* and *neural-guided solvers*. Standalone neural solvers directly address SAT instances (Bünz & Lamm, 2017; Selsam et al., 2019; Cameron et al., 2020; Shi et al., 2023). In contrast, neural-guided solvers focus on enhancing the search heuristics of classical SAT solvers (Zhang et al., 2020; Li & Si, 2022). Our CORAL paradigm leverages information from original graph problems to learn more robust and generalizable representations, thereby also improving SAT solving performance.

## 3 METHODOLOGY

In this section, we present details of our contrastive **C**ombinatorial **O**ptimization **R**epresentation **A**lignment and **L**earning (**CORAL**) paradigm. We start by introducing the preliminary background on representations of graph decision problems and SAT in Sec. 3.1. Then, we elaborate on our approach to aligning multiple problem types in Sec. 3.2. Finally, we introduce the overall pipeline and model implementation of our CORAL, as well as some important training details in Sec. 3.3.

### 3.1 PRELIMINARY

#### 3.1.1 GRAPH DECISION PROBLEM

The graph decision problem (GDP) is a fundamental computational challenge in graph theory and combinatorial optimization, where the goal is to determine the existence of specific properties within a given graph. These properties can vary widely, from identifying whether a graph contains a particular substructure, such as a clique or cycle, to assessing whether it meets conditions like connectivity or planarity. Graph decision problems are typically formulated as yes/no questions, making them essential in complexity theory, especially in the context of NP-complete problems.

ML-based models can be effectively utilized to address GDPs. In such models, the objective is to learn a representation of a specific GDP type and use it to predict decisions based on the input graph. These representations can be understood as mappings that translate the structural properties of the input graphs into corresponding decisions, thereby capturing the underlying patterns required for decision-making in GDPs.

#### 3.1.2 SAT PROBLEM

A Boolean formula in propositional logic consists of Boolean variables connected by logical operators "and" ($\wedge$), "or" ($\vee$), and "not" ($\neg$). A literal, denoted as $l_i$, is defined as either a variable or its negation, and a clause $c_j$ is represented as a disjunction of $n$ literals, $\bigvee_{i=1}^{n} l_i$. A Boolean formula is in Conjunctive Normal Form (CNF) if it is expressed as a conjunction of clauses $\bigwedge_{j=1}^{m} c_j$. Given a CNF formula, the Boolean Satisfiability Problem (SAT) aims to determine whether there exists an assignment $\pi$ of Boolean values to its variables under which the formula evaluates to true. If such an assignment $\pi$ exists, the formula is called satisfiable, where $\pi$ is called a satisfying assignment; otherwise, it is unsatisfiable. Identifying a satisfying assignment for a Boolean formula proves its satisfiability, and serves as a crucial step in solving practical instances in various applied domains. On the other hand, for an unsatisfiable formula, a minimal subset of clauses whose conjunction remains unsatisfiable is referred to as the unsat core. This subset captures the essential structure responsible for the unsatisfiability. The variables involved in this unsat core are termed unsat core variables. Identifying the unsat core variables is important for understanding the fundamental sources of unsatisfiability, and plays a critical role in optimization processes.

Graph representations play an important role in analyzing SAT formula, with four primary forms (Biere et al., 2009) commonly used: the literal-clause graph (LCG), literal-incidence graph (LIG), variable-clause graph (VCG), and variable-incidence graph (VIG). The LCG is a bipartite graph consisting of two types of nodes—literals and clauses—where an edge between a literal and a clause signifies the occurrence of that literal in the clause. The LIG, in contrast, consists solely of literal nodes, with edges representing the co-occurrence of two literals within the same clause. The VCG and VIG are derived from the LCG and LIG by merging each literal with its negation.

## 3.2 MODAL ALIGNMENT

We aim to enhance the learned representations of graph instances across a diverse range of GDPs by incorporating and synthesizing information from multiple GDP types. Specifically, we conceptualize each GDP type as a distinct problem modality. By adopting this multi-modal perspective, we explore the potential for cross-modal information-passing schemes. Note that the term 'modality' is not strictly defined. We hope to express that the problems represent different forms of a higher-level underlying difficulty and share a common underlying structure.

However, significant challenges arise due to the inherent disparities and structural gaps between different GDP types, often exhibiting varying graph topologies and problem characteristics. These differences make direct information transfer across modalities impractical and potentially detrimental to the integrity of the representations.

To address the challenges, we propose introducing SAT as a unified intermediary modality. The core concept involves transforming each GDP instance into its corresponding CNF formula, effectively con-

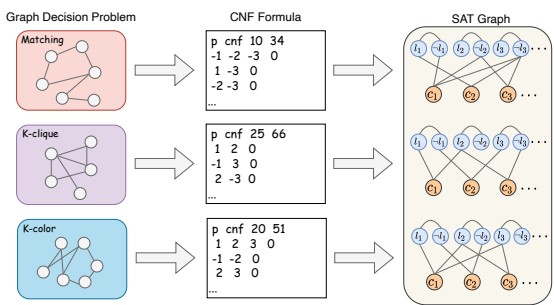

Figure 1: Transformation process from various GDP instances to the unified LCG representation of SAT.

verting it into a SAT instance. Once transformed, we construct a SAT-based graph representation for each instance, ensuring that all GDP instances, regardless of their original modalities, are standardized into an equivalent SAT graph representation. This transformation allows for uniform modeling across disparate problem types. Fig. 1 clarifies our approach to modal transformation.

After this transformation, we leverage contrastive learning to align the different modalities. Specifically, each GDP instance and its corresponding SAT instance form a positive pair, while SAT instances derived from other GDP instances within the same GDP type serve as negative samples. The SAT modality, in turn, aligns with all other modalities.

This approach facilitates effective cross-modal information transfer between GDP modalities in an indirect manner. By utilizing SAT as an intermediary modality, we preserve the distinct characteristics of each problem type while promoting coherent information fusion across modalities.

### 3.3 CORAL PARADIGM

#### 3.3.1 OVERVIEW

In this section, we provide a detailed introduction to CORAL. Fig. 2 exhibits an overview.

Consider a scenario involving $n$ types of GDPs, denoted as $\mathcal{P}_1, \mathcal{P}_2, \ldots, \mathcal{P}_n$, along with $n$ corresponding graph sets $\mathbf{G}_1, \mathbf{G}_2, \ldots, \mathbf{G}_n$. For simplicity, assume that each graph set $\mathbf{G}_i$ contains $m$ graphs, i.e., $\mathbf{G}_i = \{\mathcal{G}_i^1, \mathcal{G}_i^2, \ldots, \mathcal{G}_i^m\}$, for $i = 1, 2, \ldots, n$. The objective is to solve problem $\mathcal{P}_i$ on graphs in $\mathbf{G}_i$. In total, there are $m \times n$ instances, denoted by $I_i^j = (\mathcal{P}_i, \mathcal{G}_j)$, where $i = 1, 2, \ldots, n$ and $j = 1, 2, \ldots, m$.

We first transform each of the $m \times n$ GDP instances into CNF, thereby generating their corresponding SAT graphs, i.e., $(\mathcal{P}_i, \mathcal{G}_j) \to \mathcal{B}_i^j$, where $\mathcal{B}_i^j$ is the constructed (bipartite) SAT graph.

Then, we develop $n$ distinct graph models, $\mathbb{M}_1, \ldots, \mathbb{M}_n$, each for one GDP type, and one unified SAT model $\mathbb{M}_{sat}$ to address the problem space. Both the graph models and the SAT model are structured around two key components: the **Representation Extractor** and the **Output Module**. The Representation Extractor is responsible for learning and extracting representations from the input graph instances, whether derived from GDP or SAT transformations. The Output Module then utilizes these learned representations to produce task-specific outputs, thereby enabling the resolution of the given problem.

In the training phase, we simultaneously train the $n + 1$ models corresponding to the $n$ GDP modalities and the SAT modality. The supervision is derived from two parts: the decision loss and the contrastive loss. The decision loss is applied independently to each model, guiding it to effectively

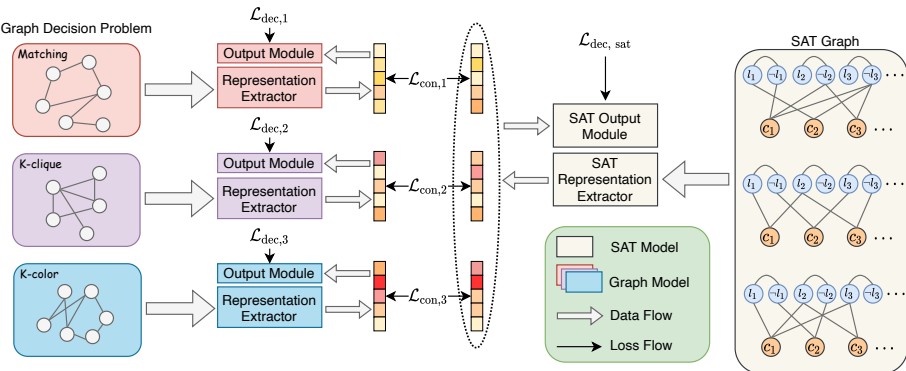

Figure 2: Overview of our CORAL paradigm. Given instances from multiple GDP types and their corresponding SAT graphs, a graph model is trained for each GDP type alongside a SAT model. Each model is composed of a Representation Extractor and an Output Module. The input graphs are processed by the Representation Extractor to generate instance-level representations, which are subsequently fed into the Output Module to produce the final decisions for each instance. The decision loss is applied individually to each model, while the contrastive loss is applied to each graph model. All contrastive losses are applied to the SAT model.

learn the feature representations of the respective instances and capture the unique characteristics of its assigned modality. Meanwhile, the contrastive loss is employed to facilitate feature fusion and message passing across the different modalities, enabling the models to leverage complementary information from multiple modalities.

### 3.3.2 MODEL ARCHITECTURE

In this section, we illustrate the utilized model architecture, encompassing both the graph and the SAT models. Please refer to Appendix B for more details.

**Graph Model.** Each graph model is designed to address a specific type of GDP, and all models maintain a consistent architecture. To illustrate this, we focus on problem $\mathcal{P}_n$ and its corresponding graph model $\mathbb{M}_n$. The graph model $\mathbb{M}_n$ takes graphs in the set $\mathbf{G}_n$ as input and processes them through the Representation Extractor. The input graph primarily consists of edge information, which is often a critical aspect of GDPs. For the initial vertex features, we introduce a $d$-dimensional embedding for all vertices, represented as $\mathbf{h}_n^{(0)}$.

For the Representation Extractor, we adopt the vanilla Graph Convolutional Network (GCN) (Kipf & Welling, 2016), which is widely used as a backbone for node embeddings in graph-based tasks. Assume there are $k$ layers, the embedding extraction at the $i$-th layer of the network is expressed as:

$$\mathbf{H}_n^{(i)} = \text{ReLU}(\tilde{\mathbf{D}}^{-\frac{1}{2}} \tilde{\mathbf{A}} \tilde{\mathbf{D}}^{-\frac{1}{2}} \mathbf{H}_n^{(i-1)} \mathbf{W}_n^{(i-1)}), \ i = 1, 2, \ldots, k, \tag{1}$$

where $\mathbf{H}$ denotes the node embedding matrix, with each row corresponding to a node embedding. The matrix $\tilde{\mathbf{A}} = \mathbf{A} + \mathbf{I}$ is the adjacency matrix augmented with self-loops through the identity matrix $\mathbf{I}$. $\tilde{\mathbf{D}}_{ii} = \sum_j \tilde{\mathbf{A}}_{ij}$ is the degree matrix, and $\mathbf{W}$ is the learnable weight matrix. Following the extraction of node features, we apply average pooling to the node embedding matrix $\mathbf{H}_n^{(k)}$ to aggregate the node-level information into a single representation for the entire graph instance, denoted as $\mathbf{r}_n$. This aggregation is computed as follows:

$$\mathbf{r}_n = \frac{\sum_{v \in \mathcal{V}} \mathbf{h}_{n,v}^{(k)}}{|\mathcal{V}|}, \tag{2}$$

where $\mathcal{V}$ represents the set of vertices in the input graph, $|\mathcal{V}|$ denotes the total number of vertices, and $\mathbf{h}_{n,v}^{(k)}$ is the extracted embedding for node $v$. $\mathbf{r}_n$ serves as the instance-level feature representation, and is subsequently fed into the Output Module, which is implemented as an MLP to produce the final decision for the instance.

**SAT Model.** Apart from the graph models, the SAT model $\mathbb{M}_{sat}$ processes the constructed SAT graphs via its own Representation Extractor. For illustration, we consider the LCG representation. For the initial node features, we define two distinct $d$-dimensional embeddings: $\mathbf{h}_l^{(0)}$ for all literal nodes and $\mathbf{h}_c^{(0)}$ for all clause nodes.

The architecture of the Representation Extractor is inspired by NeuroSAT (Selsam et al., 2019). For notational clarity, we assume that the extractor consists of $k$ layers, with both literal and clause node embeddings being iteratively aggregated and updated at each layer. At the $i$-th layer, the updates for the literal and clause node embeddings are formulated as follows:

$$\mathbf{h}_l^{(i)} = \text{LayerNormLSTM}\left(\underset{c \in \mathcal{N}(l)}{\text{SUM}}\left(\text{MLP}\left(\mathbf{h}_c^{i-1}\right)\right), \mathbf{h}_l^{(i-1)}, \mathbf{h}_{\neg l}^{(i-1)}\right), \tag{3}$$

$$\mathbf{h}_c^{(i)} = \text{LayerNormLSTM}\left(\underset{l \in \mathcal{N}(c)}{\text{SUM}}\left(\text{MLP}\left(\mathbf{h}_l^{i-1}\right)\right), \mathbf{h}_c^{(i-1)}\right), \tag{4}$$

where $l$ and $c$ represent an arbitrary literal node and clause node, respectively, $\mathcal{N}(\cdot)$ refers to the set of neighboring nodes. The summation operator (SUM) serves as the aggregation function, while LayerNormLSTM (Ba, 2016) is employed as the update function.

Similar to the graph models, the instance-level representation $\mathbf{r}_{sat}$ derives by averaging the literal node embeddings after the $k$-th layer. The instance-level representation, along with the literal-level embeddings, is passed to the Output Module, which is also implemented as an MLP, to generate the final task-specific decisions or predictions.

### 3.3.3 LOSS FUNCTION

In CORAL paradigm, model training is guided by two key loss functions: the decision loss and the contrastive loss. These losses play a critical role in optimizing the model's performance, with the decision loss focusing on task-specific predictions, while the contrastive loss facilitates cross-modal representation alignment and feature fusion.

**The Decision Loss.** The decision loss $\mathcal{L}_{\text{dec}}$ is defined as a binary cross-entropy loss, which can be computed by:

$$\mathcal{L}_{\text{dec}} = \sum_{i \in \text{Batch}} \left\{-d_i^{\text{gt}} \log(d_i^{\text{out}}) - (1 - d_i^{\text{gt}}) \log(1 - d_i^{\text{out}})\right\}, \tag{5}$$

where $d^{\text{out}}$ denotes the output decision of the models, and $d^{\text{gt}}$ refers to the ground truth label for satisfiability. For each model, the decision loss is independently computed and applied.

**The Contrastive Loss.** Inspired by Chen et al. (2020b), we define the contrastive loss $\mathcal{L}_{\text{con}}$ to facilitate the alignment between the GDP and SAT modalities. Taking $\mathcal{P}_n$ and the SAT modality as an example, the contrastive loss is formulated as follows:

$$\mathcal{L}_{\text{con},n} = \sum_{i=1}^{N} \left\{ -\log \frac{\exp(sim(\hat{\mathbf{r}}_n^i, \hat{\mathbf{r}}_{sat}^i)/\tau)}{\sum_{j=1}^{N} \mathbb{I}_{j \neq i} \exp(sim(\hat{\mathbf{r}}_n^i, \hat{\mathbf{r}}_{sat}^j)/\tau)} - \log \frac{\exp(sim(\hat{\mathbf{r}}_n^i, \hat{\mathbf{r}}_{sat}^i)/\tau)}{\sum_{j=1}^{N} \mathbb{I}_{j \neq i} \exp(sim(\hat{\mathbf{r}}_n^j, \hat{\mathbf{r}}_{sat}^i)/\tau)} \right\} \tag{6}$$

where $N$ represents the number of instance pairs in a batch, $\hat{\mathbf{r}}_n^i$ denotes the normalized representation of the $i$-th instance in the $\mathcal{P}_n$ modality, and $\hat{\mathbf{r}}_{sat}^i$ denotes the normalized representation of the corresponding instance in the SAT modality, derived from the $i$-th instance of the $\mathcal{P}_n$ modality. The parameter $\tau$ is the temperature scalar, and $\mathbb{I}$ is an indicator function. The function $sim(\cdot, \cdot)$ measures the cosine similarity between two representations, defined as $sim(\mathbf{r}_i, \mathbf{r}_j) = \frac{\mathbf{r}_i^\top \mathbf{r}_j}{\|\mathbf{r}_i\|\|\mathbf{r}_j\|}$.

Each GDP modality is trained using the contrastive loss with the SAT modality, allowing independent optimization for each GDP model. In parallel, the SAT model is optimized using the average contrastive losses computed across all GDP modalities, ensuring effective alignment.

### 3.3.4 TRAINING DETAILS

We adopt a warm start strategy to ensure the models learn robust representations. During the initial training phase, only the decision loss is utilized, while the contrastive loss is temporarily disabled.

This phase allows the models to focus on learning meaningful task-specific representations based solely on the decision outcomes. Our insight is to provide a stable foundation for representation learning before introducing the more complex cross-modal alignment enforced by contrastive loss.

After the warm start phase, we introduce the contrastive loss alongside the decision loss. To balance the influence of these two losses, we introduce a parameter $\beta$, which controls the relative weight of the decision loss during the joint training phase.

It is important to note that CORAL serves solely as a pre-training framework. While it enables the learning of robust and transferable representations, fine-tuning is required for optimal performance.

## 4 EXPERIMENTS

### 4.1 EXPERIMENTAL SETUP

**Datasets.**  To evaluate the broad applicability of our approach, we select seven GDPs: $k$-Clique, $k$-Dominating Set ($k$-Domset), $k$-Vertex Cover ($k$-Vercov), $k$-Coloring ($k$-Color), $k$-Independent Set ($k$-Indset), Perfect Matching (Matching), and Graph Automorphism (Automorph). For each problem, we randomly generate graph instances that adhere to a distribution specific to the problem. To ensure a comprehensive and rigorous evaluation, we create datasets with varying levels of difficulty, categorized as easy, medium, and hard, based on the size and distribution of the generated graphs. For each easy and medium dataset, we generate 160,000 instances for training, 20,000 instances for validation, and 20,000 instances for testing. For each hard dataset, we only produce 20,000 instances for testing to evaluate the generalizability of models. Additionally, we ensure an equal distribution of labels, with 50% of instances labeled as satisfiable (1) and 50% as unsatisfiable (0) across the training, validation, and test sets. The graph instances were transformed into CNF using generators from CNFGen (Lauria et al., 2017). Moreover, we synthetically generate instances of two pseudo-industrial SAT problems, employing the Community Attachment (CA) model (Giráldez-Cru & Levy, 2015) and the Popularity-Similarity (PS) model (Giráldez-Cru & Levy, 2017), and two random SAT problems, utilizing the SR generator in NeuroSAT (Selsam et al., 2019) and the 3-SAT generator in CNFGen (Lauria et al., 2017), to demonstrate the effectiveness of the learned representations on unseen datasets, thereby proving that the representations are high-level. Please refer to Appendix A for more details about the datasets.

**Tasks.**  We evaluate the performance of our graph models on the **GDP solving** task, focusing on their ability to accurately determine the solution for each specific problem type. For the SAT model, we assess its effectiveness on the **satisfiability prediction** task. Moreover, we further evaluate the SAT model on two essential tasks critical to SAT solving: **satisfying assignment prediction** and **unsat core variable prediction**. Satisfying assignment prediction requires the model to determine a specific variable assignment that satisfies the given SAT instance, while unsat core variable prediction involves identifying the minimal subset of variables that contribute to the unsatisfiability of the instance. These tasks are crucial for evaluating the generalizability of the learned representations.

**Baselines.**  For a fair comparison, we establish baselines for both the graph and SAT models. The baseline for our graph models consists of models with the same architecture as our proposed approach but trained in a conventional manner, without leveraging the contrastive learning framework. Each graph model is trained independently on its respective dataset using standard supervised learning. Similarly, the baseline for our SAT model adopts the same architecture as in our proposed method but is trained simultaneously on seven GDP datasets in a traditional manner, without cross-modal contrastive alignment.

### 4.2 GRAPH MODEL PERFORMANCE

#### 4.2.1 GDP SOLVING

We evaluate the accuracy of the graph models in solving seven GDPs. The baseline model, denoted as **Graph Model**, follows the architecture outlined in Sec. 3.3.2 and is trained independently on each of the seven GDP datasets using conventional supervised learning. Our proposed approach, denoted as **Graph Model+Contrast**, employs the same architecture as the baseline model but initializes model parameters with a pre-trained checkpoint from CORAL, trained on the seven GDP datasets, and is then fine-tuned individually on the seven GDP datasets.

Table 1: GDP solving accuracy of the graph models trained on identical distribution. The 'Overall' column represents the average accuracy across all datasets.

| Difficulty | Model | k-Clique | k-Domset | k-Vercov | k-Color | k-Indset | Matching | Automorph | Overall |
|---|---|---|---|---|---|---|---|---|---|
| Easy | Graph Model | 0.770 | 0.585 | 0.603 | 0.861 | 0.627 | 0.712 | 0.636 | 0.685 |
| | Graph Model+Contrast | **0.793** | **0.620** | **0.673** | **0.902** | **0.675** | **0.717** | **0.654** | **0.719** |
| Medium | Graph Model | 0.632 | 0.622 | 0.599 | 0.796 | 0.611 | 0.706 | 0.633 | 0.657 |
| | Graph Model+Contrast | **0.713** | **0.646** | **0.633** | **0.822** | **0.640** | **0.728** | **0.657** | **0.691** |

Table 1 presents the results, showing the performance of both models trained and evaluated on datasets with identical distributions, including the easy and medium datasets. Our approach consistently outperforms the baseline model across multiple GDP tasks, indicating that leveraging the pre-trained representations from CORAL significantly enhances the models' ability to solve various GDPs. Furthermore, it supports our motivation that the high-level representations learned by CORAL enable mutual enhancement, where insights and patterns learned from one problem type can be transferred to and improve the solution of others.

### 4.2.2 Generalization on Hard Datasets

To assess the generalization capabilities of the graph models, we evaluate their performance on the hard datasets, which consist of problem instances with increased scale and complexity. The model names and training configurations are consistent with those described in Sec. 4.2.1.

Table 2: GDP solving accuracy of the graph models on the hard datasets. The terms 'Easy' and 'Medium' in parentheses indicate the difficulty level of the datasets used for training. The 'Overall' column represents the average accuracy across all datasets.

| Model | k-Clique | k-Domset | k-Vercov | k-Color | k-Indset | Matching | Automorph | Overall |
|---|---|---|---|---|---|---|---|---|
| Graph Model (Easy) | 0.545 | 0.500 | 0.500 | 0.546 | **0.505** | 0.664 | 0.631 | 0.556 |
| Graph Model+Contrast (Easy) | **0.571** | **0.501** | 0.500 | **0.605** | 0.503 | **0.679** | **0.636** | **0.571** |
| Graph Model (Medium) | 0.571 | 0.562 | 0.500 | 0.637 | 0.531 | 0.683 | 0.632 | 0.588 |
| Graph Model+Contrast (Medium) | **0.578** | **0.565** | **0.577** | **0.676** | **0.565** | **0.700** | **0.653** | **0.616** |

Table 2 presents the results of graph models trained on the easy and medium datasets, and tested on the hard datasets. The results clearly show that models leveraging the pre-trained representations from CORAL exhibit improved performance across most GDP tasks, indicating that CORAL not only enhances task-specific performance but also provides robust generalization to more challenging and previously unseen problem instances. The consistent improvements highlight the ability of CORAL to capture and leverage the inherent connections among different CO problems to learn representations that transcend individual problem types.

### 4.3 SAT Model Performance

### 4.3.1 Satisfiability Prediction

**Satisfiability Prediction Accuracy.** We assess the satisfiability prediction accuracy of the SAT model using instances transformed from seven distinct GDPs. The baseline model, referred to as the **SAT Model**, adheres to the architecture described in Sec. 3.3.2 and is trained concurrently on instances derived from all seven GDPs. This training strategy capitalizes on the relatively coherent graph representations of the SAT instances. Our proposed approach, denoted as **SAT Model+Contrast**, employs the same architecture as the baseline model, but the model parameters are initialized with a pre-trained checkpoint from CORAL, trained on the seven GDP datasets. The model is then fine-tuned on the instances transformed from all seven GDPs simultaneously.

Table 3: Satisfiability prediction accuracy of the SAT models. The 'Overall' column represents the average accuracy across all datasets.

| Difficulty | Model | k-Clique | k-Domset | k-Vercov | k-Color | k-Indset | Matching | Automorph | Overall |
|---|---|---|---|---|---|---|---|---|---|
| Easy | SAT Model | 0.959 | 0.991 | 0.998 | 0.974 | 0.954 | 0.995 | 0.999 | 0.981 |
| | SAT Model+Contrast | **0.989** | **0.996** | **0.999** | **0.988** | **0.989** | **0.999** | 0.999 | **0.994** |
| Medium | SAT Model | 0.876 | 0.987 | 0.991 | 0.817 | 0.887 | 0.997 | 0.988 | 0.935 |
| | SAT Model+Contrast | **0.923** | **0.991** | **0.996** | **0.946** | **0.930** | **0.999** | **0.999** | **0.969** |

Table 3 shows the results, where our approach consistently outperforms the baseline model on most datasets, with particularly notable improvements on more challenging datasets. The results demonstrate the effectiveness of leveraging the inherent connections between different CO problems. By drawing on the common underlying characteristics among different problem types, our approach enhances the performance of the SAT model, showcasing the advantages of cross-domain learning.

**Generalization Performance.** We evaluate the generalization capabilities of the SAT models on instances transformed from hard GDP datasets, with the results presented in Table 4. Our proposed approach consistently outperforms the baseline model across most datasets, underscoring the robustness and transferability of the representations learned through CORAL, and its ability to generalize across complex, unseen problem instances.

Table 4: Satisfiability prediction accuracy of the SAT models on the hard datasets. The terms 'Easy' and 'Medium' in parentheses indicate the difficulty level of the datasets used for training. The 'Overall' column represents the average accuracy across all datasets.

| Model | k-Clique | k-Domset | k-Vercov | k-Color | k-Indset | Matching | Automorph | Overall |
|---|---|---|---|---|---|---|---|---|
| SAT Model (Easy) | 0.475 | 0.505 | 0.500 | 0.588 | 0.473 | 0.995 | 0.729 | 0.609 |
| SAT Model+Contrast (Easy) | **0.662** | **0.506** | 0.500 | **0.600** | **0.665** | **0.998** | **0.790** | **0.674** |
| SAT Model (Medium) | 0.692 | 0.964 | 0.852 | 0.679 | 0.694 | 0.996 | 0.990 | 0.838 |
| SAT Model+Contrast (Medium) | **0.827** | **0.972** | **0.936** | **0.745** | **0.836** | **0.997** | **0.991** | **0.901** |

### 4.3.2 OTHER SAT-BASED TASKS

We further evaluate the SAT model on the satisfying assignment prediction task and the unsat core variable prediction task. To assess performance, we compare three different approaches by tracking the accuracy over training iterations. For our proposed approach, referred to as **SAT Model+Contrast**, we initialize the model using a pre-trained checkpoint obtained from CORAL, trained on the seven GDP datasets, and subsequently fine-tune it on individual datasets. For comparison, we include two baseline models: **SAT Model**, which is initialized with a pre-trained checkpoint trained in a conventional manner on the seven GDP datasets, and **Un-Pretrained SAT Model**, which is trained from scratch. The results are shown in Fig. 3.

On the datasets encountered during pre-training, both our approach and the pre-trained baseline significantly outperform the un-pretrained baseline. However, our approach demonstrates superior performance by achieving faster convergence and attaining a higher final accuracy. On the unseen datasets, our approach still outperforms the baseline models, whereas the pre-trained and un-pretrained baselines exhibit comparable performance. These results highlight the effectiveness of the CORAL paradigm, which not only improves convergence rates but also enhances the model's ability to generalize to previously unseen datasets, thereby demonstrating the strength of leveraging contrastive learning across multiple problem types. Please refer to Appendix D for more results.

The above results collectively validate the efficacy of CORAL, demonstrating its capacity to enhance both in-domain performance and cross-domain generalization.

### 4.4 EXPERIMENTS ON MORE BACKBONES

We conduct experiments on more backbones to show the consistent effectiveness of CORAL. We consider LCG and VCG modeling for the SAT graphs. We also employ alternative backbones, including GCN for the SAT model and GraphSAGE (Hamilton et al., 2017a) for the graph models. Table 5 shows the results, where our approaches consistently outperform the baselines. In particular, when employing the GraphSAGE backbone, our approach achieves a remarkable improvement over the baseline. Please refer to Appendix B for more details on model backbones.

## 5 CONCLUSION AND OUTLOOK

In this paper, we introduce CORAL, a novel paradigm designed to promote learning representation for CO problems via contrastive learning across different problem types. By focusing on graph decision problems and leveraging the inherent connections, CORAL effectively captures the shared structural characteristics across different problem types. We perform extensive experiments on multiple datasets and tasks. The results indicate that our approach shows not only improved task-specific

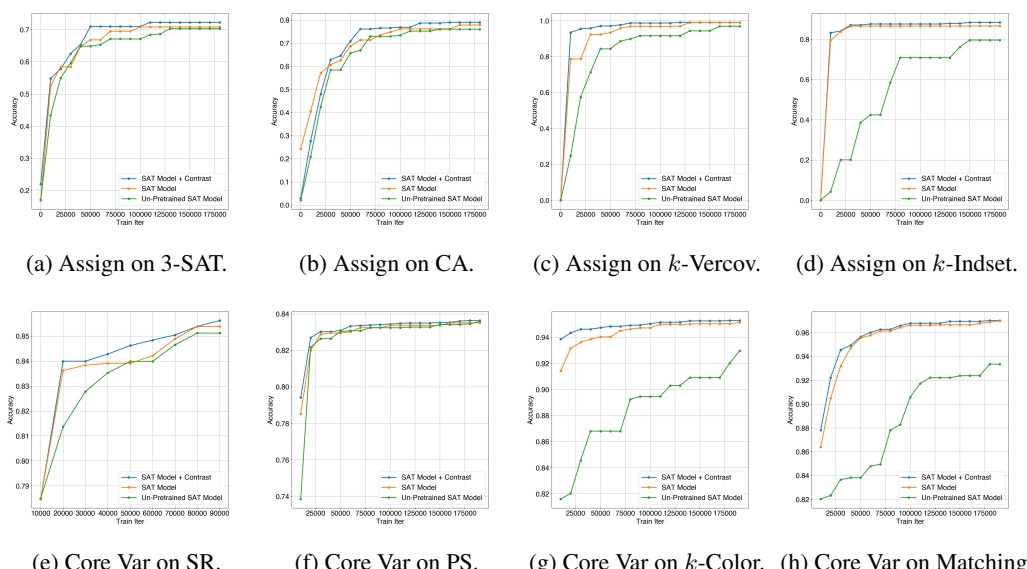

| (a) Assign on 3-SAT. | (b) Assign on CA. | (c) Assign on $k$-Vercov. | (d) Assign on $k$-Indset. |
|---|---|---|---|

| (e) Core Var on SR. | (f) Core Var on PS. | (g) Core Var on $k$-Color. | (h) Core Var on Matching. |
|---|---|---|---|

Figure 3: Model performance w.r.t. training iterations on SAT-based tasks across various datasets. The top four graphs display the results for the satisfying assignment prediction task (Assign), while the bottom four graphs present the results for the unsat core variable prediction task (Core Var). The left four graphs depict the model's performance on unseen datasets, whereas the right four graphs illustrate the performance on datasets encountered during the pre-training phase.

Table 5: Experimental results across various model backbones. The table presents the GDP-solving accuracy for the graph models and the satisfiability prediction accuracy for the SAT models. 'SAT Back.' refers to SAT model backbone, and 'Graph Back.' denotes graph model backbone.

| SAT Back. | Graph Back. | Difficulty | Model | k-Clique | k-Domset | k-Vercov | k-Color | k-Indset | Matching | Automorph | Overall |
|---|---|---|---|---|---|---|---|---|---|---|---|
| LCG+GCN | GCN | Easy | Graph Model | 0.770 | 0.585 | 0.603 | 0.861 | 0.627 | **0.712** | 0.636 | 0.685 |
| | | | Graph Model+Contrast | **0.793** | **0.611** | **0.650** | **0.896** | **0.677** | 0.711 | **0.646** | **0.712** |
| | | Medium | Graph Model | 0.632 | 0.622 | 0.599 | 0.796 | 0.611 | 0.706 | 0.633 | 0.657 |
| | | | Graph Model+Contrast | **0.715** | **0.654** | **0.634** | **0.817** | **0.640** | **0.723** | **0.644** | **0.690** |
| | | Easy | SAT Model | 0.763 | 0.790 | 0.890 | 0.868 | 0.780 | 0.801 | 0.616 | 0.787 |
| | | | SAT Model+Contrast | **0.827** | **0.932** | **0.953** | **0.937** | **0.820** | **0.967** | **0.689** | **0.875** |
| | | Medium | SAT Model | 0.724 | 0.652 | 0.836 | 0.858 | 0.721 | 0.835 | 0.668 | 0.756 |
| | | | SAT Model+Contrast | **0.752** | **0.953** | **0.979** | **0.887** | **0.748** | **0.994** | **0.784** | **0.871** |
| VCG+GCN | GCN | Easy | Graph Model | 0.770 | 0.585 | 0.603 | 0.861 | 0.627 | **0.712** | 0.636 | 0.685 |
| | | | Graph Model+Contrast | **0.780** | **0.606** | **0.629** | **0.888** | **0.663** | 0.711 | **0.642** | **0.703** |
| | | Medium | Graph Model | 0.632 | 0.622 | 0.599 | 0.796 | 0.611 | 0.706 | 0.633 | 0.657 |
| | | | Graph Model+Contrast | **0.708** | **0.642** | **0.630** | **0.804** | **0.621** | **0.718** | **0.640** | **0.680** |
| | | Easy | SAT Model | 0.511 | 0.840 | 0.919 | 0.828 | 0.491 | 0.813 | 0.568 | 0.710 |
| | | | SAT Model+Contrast | **0.809** | **0.959** | **0.993** | **0.947** | **0.795** | **0.993** | **0.744** | **0.891** |
| | | Medium | SAT Model | 0.669 | 0.946 | 0.950 | 0.860 | 0.677 | 0.988 | 0.642 | 0.819 |
| | | | SAT Model+Contrast | **0.748** | **0.988** | **0.995** | **0.898** | **0.745** | **0.994** | **0.734** | **0.872** |
| LCG+NeuroSAT | GraphSAGE | Easy | Graph Model | 0.579 | 0.500 | 0.507 | 0.618 | 0.522 | 0.582 | 0.538 | 0.549 |
| | | | Graph Model+Contrast | **0.797** | **0.632** | **0.708** | **0.933** | **0.753** | **0.710** | **0.639** | **0.739** |
| | | Medium | Graph Model | 0.528 | 0.565 | 0.560 | 0.552 | 0.500 | 0.582 | 0.548 | 0.548 |
| | | | Graph Model+Contrast | **0.728** | **0.641** | **0.667** | **0.859** | **0.701** | **0.717** | **0.648** | **0.709** |
| | | Easy | SAT Model | 0.959 | 0.991 | 0.998 | 0.974 | 0.954 | 0.995 | 0.999 | 0.981 |
| | | | SAT Model+Contrast | **0.990** | **0.996** | **0.999** | **0.988** | **0.991** | **0.999** | 0.999 | **0.995** |
| | | Medium | SAT Model | 0.876 | 0.987 | 0.991 | 0.817 | 0.887 | 0.997 | 0.988 | 0.935 |
| | | | SAT Model+Contrast | **0.925** | **0.991** | **0.996** | **0.953** | **0.935** | **0.999** | **0.997** | **0.971** |

performance but also robust generalization capabilities to more complex and unseen instances and problems, underscoring the potential of CORAL as a unified pre-training paradigm for CO research.

Our future work will focus on addressing the current limitations. First, we aim to explore unsupervised learning approaches to minimize dependence on labeled data, thereby enhancing applicability and scalability in data-sparse scenarios. Additionally, we will focus on developing more generalized unifying approaches to bridge various CO problems, making the learning process more accessible and applicable across a broader range of problem domains.

ETHICS STATEMENT

This work adheres to ethical standards in research and does not involve any direct human subjects, nor does it present any privacy or security concerns. The datasets used in this study are synthetically generated without involving sensitive or personally identifiable information. All experiments and methodologies were conducted in compliance with legal regulations and established research integrity practices. There are no known conflicts of interest, sponsorship influences, or concerns related to discrimination, bias, or fairness in our approach. Additionally, the research does not produce any harmful insights or applications, and efforts have been made to ensure that the work promotes the advancement of combinatorial optimization without negative societal impact.

REPRODUCIBILITY STATEMENT

We have made efforts to ensure the reproducibility of the results presented in this paper. The architectural details of our models, including the graph models and SAT models, are described in Sec. 3.3.2. We show more details in Appendix B. The loss functions are illustrated in Sec. 3.3.3 and Appendix C. Furthermore, the datasets used for experiments are detailed in the Appendix A, with all relevant settings provided to ensure consistency across experiments. Important training parameters are shown in Appendix D. We will release our source code once the paper is accepted.

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

APPENDIX

# A    MORE DETAILS ON DATASETS

In this section, we supplement more details on the utilized datasets in our main paper, including the parameters of GDP instances and the statistics of SAT instances.

## A.1    GDP INSTANCES

To ensure the generation of high-quality GDP instances that accurately capture the inherent characteristics of each problem, we carefully select the graph distributions and parameters used for instance generation. Some parameters refer to Li et al. (2023). Table 6 provides a detailed overview of the specific GDP datasets employed in the main paper.

Table 6: Details of generated GDP datasets.

| Dataset | Description | Parameters | Notes |
|---------|-------------|------------|-------|
| $k$-Clique | The $k$-Clique dataset consists of graph instances of the $k$-Clique problem, which involves determining whether a given graph contains a clique of size $k$. A clique is a subset of vertices in which every pair of vertices is connected by an edge. The goal is to identify whether such a fully connected subset of $k$ vertices exists within the graph. Instances are built on randomly generated Erdős-Rényi graphs. Parameters include number of vertices $v$, edge probabilities $p$, and clique size $k$. | General: $p = \binom{v}{k}^{-1/\binom{v}{2}}$, Easy dataset: $v \sim \text{Uniform}(5, 15)$, $k \sim \text{Uniform}(3, 4)$, Medium dataset: $v \sim \text{Uniform}(15, 20)$, $k \sim \text{Uniform}(3, 5)$, Hard dataset: $v \sim \text{Uniform}(20, 25)$, $k \sim \text{Uniform}(4, 6)$. | The parameter $p$ is selected based on Bollobás & Erdös (1976), ensuring that the expected number of $k$-cliques in the generated graph is equal to 1. |
| $k$-Domset | The $k$-Domset dataset consists of graph instances of the $k$-Dominating Set problem, which involves determining whether a given graph contains a dominating set of size $k$. A dominating set is a subset of vertices such that every vertex in the graph is either in the subset or adjacent to at least one vertex in the subset. The goal is to identify whether such a subset of $k$ vertices exists that can 'dominate' the entire graph, ensuring that all other vertices are either in the subset or connected to it. Instances are built on randomly generated Erdős-Rényi graphs. Parameters include number of vertices $v$, edge probabilities $p$, and dominating set size $k$. | General: $p = 1 - \left(1 - \binom{v}{k}^{-1/(v-k)}\right)^{1/k}$, Easy dataset: $v \sim \text{Uniform}(5, 15)$, $k \sim \text{Uniform}(2, 3)$, Medium dataset: $v \sim \text{Uniform}(15, 20)$, $k \sim \text{Uniform}(3, 5)$, Hard dataset: $v \sim \text{Uniform}(20, 25)$, $k \sim \text{Uniform}(4, 6)$. | The parameter $p$ is selected based on Wieland & Godbole (2001), ensuring that the expected number of $k$-dominating sets in the generated graph is equal to 1. |
| $k$-Vercov | The $k$-Vercov dataset consists of graph instances of the $k$-Vertex Cover problem, which involves determining whether a given graph contains a vertex cover of size $k$. A vertex cover is a subset of vertices such that every edge in the graph is incident to at least one vertex in the subset. The goal is to identify whether a subset of $k$ vertices exists that can 'cover' all the edges in the graph, ensuring that each edge is connected to at least one vertex in the subset. Instances are built on randomly generated Erdős-Rényi graphs. Parameters include number of vertices $v$, edge probabilities $p$, and vertex set size $k$. | General: $p = \binom{v}{k}^{-1/\binom{v}{2}}$, Easy dataset: $v \sim \text{Uniform}(5, 15)$, $k \sim \text{Uniform}(3, 5)$, Medium dataset: $v \sim \text{Uniform}(10, 20)$, $k \sim \text{Uniform}(6, 8)$, Hard dataset: $v \sim \text{Uniform}(15, 25)$, $k \sim \text{Uniform}(9, 10)$. | The parameter $p$ is selected based on the relationship between $k$-Clique and $k$-Vercov, ensuring that the expected size of the minimum vertex cover in the generated graph is $k$. |
| $k$-Color | The $k$-Color dataset consists of graph instances of the $k$-Coloring problem, which involves determining whether a given graph can be colored with $k$ colors such that no two adjacent vertices share the same color. A valid coloring assigns one of $k$ different colors to each vertex, ensuring that vertices connected by an edge have different colors. The goal is to identify whether such a coloring scheme exists for the graph using at most $k$ colors. Instances are built on randomly generated Erdős-Rényi graphs. Parameters include number of vertices $v$, edge probabilities $p$, and number of colors $k$. | General: $p = \binom{v}{k}^{-1/\binom{v}{2}}$, Easy dataset: $v \sim \text{Uniform}(5, 15)$, $k \sim \text{Uniform}(3, 4)$, Medium dataset: $v \sim \text{Uniform}(15, 20)$, $k \sim \text{Uniform}(3, 5)$, Hard dataset: $v \sim \text{Uniform}(20, 25)$, $k \sim \text{Uniform}(4, 6)$. | The parameter $p$ is selected based on the relationship between $k$-Clique and $k$-Color, ensuring that the expected minimum number of colors for the generated graph is $k$. |
| $k$-Indset | The $k$-Indset dataset consists of graph instances of the $k$-Independent Set problem, which involves determining whether a given graph contains an independent set of size $k$. An independent set is a subset of vertices in which no two vertices are adjacent, meaning there are no edges connecting any pair of vertices in the subset. The goal is to identify whether such a subset of $k$ vertices exists within the graph, ensuring that the selected vertices are mutually non-adjacent. Instances are built on randomly generated Erdős-Rényi graphs. Parameters include number of vertices $v$, edge probabilities $p$, and independent set size $k$. | General: $p = 1 - \binom{v}{k}^{-1/\binom{v}{2}}$, Easy dataset: $v \sim \text{Uniform}(5, 15)$, $k \sim \text{Uniform}(3, 4)$, Medium dataset: $v \sim \text{Uniform}(15, 20)$, $k \sim \text{Uniform}(3, 5)$, Hard dataset: $v \sim \text{Uniform}(20, 25)$, $k \sim \text{Uniform}(4, 6)$. | The parameter $p$ is selected based on the relationship between $k$-Clique and $k$-Indset, ensuring that the expected number of $k$-independent sets in the generated graph is equal to 1. |
| Matching | The Matching dataset consists of graph instances of the Perfect Matching problem, which involves determining whether a given graph contains a perfect matching. A perfect matching is a subset of edges in which every vertex is incident to exactly one edge in the subset. In other words, the graph's vertices can be paired off so that no vertex is left unpaired and no two edges share a vertex. The goal is to identify whether such a perfect matching exists within the graph, ensuring that all vertices are perfectly matched. Instances are built on randomly generated Erdős-Rényi graphs. Parameters include number of vertices $v$ and edge probabilities $p$. | General: $p = \ln(v)/v$, Easy dataset: $v \sim \text{Uniform}(6, 16)$, should be an even number, Medium dataset: $v \sim \text{Uniform}(16, 24)$, should be an even number, Hard dataset: $v \sim \text{Uniform}(24, 30)$, should be an even number. | The selected parameter $p$ is a sharp threshold for graph connectivity based on Erdos et al. (1960), ensuring that the generated graph is neither too dense nor too sparse. |
| Automorph | The Automorph dataset consists of graph instances of the Graph Automorphism problem, which involves determining whether a given graph has a non-trivial automorphism. An automorphism is a mapping of the graph's vertices to itself such that the structure of the graph is preserved, meaning that the adjacency relationships between vertices remain unchanged. The goal is to identify whether there exists a way to rearrange the vertices of the graph such that it appears identical to its original form. Instances are built on randomly generated Erdős-Rényi graphs. Parameters include number of vertices $v$ and edge probabilities $p$. | General: $p = \ln(v)/v$, Easy dataset: $v \sim \text{Uniform}(4, 8)$, Medium dataset: $v \sim \text{Uniform}(8, 10)$, Hard dataset: $v \sim \text{Uniform}(10, 12)$. | The selected parameter $p$ is a sharp threshold for graph connectivity based on Erdos et al. (1960), ensuring that the generated graph is neither too dense nor too sparse. |

Note that six of the seven GDPs are NP-hard, while the Perfect Matching problem is a P problem.

## A.2 SAT INSTANCES

After generating the seven GDP datasets, the corresponding seven SAT datasets are generated by transforming the GDP datasets, utilizing the python toolkit CNFGen (Lauria et al., 2017). We also compute the statistics of those SAT datasets to provide comprehensive information on datasets. The dataset statistics are shown in Table 7.

Table 7: SAT dataset statistics. # Variables refers to average number of variables, # Clauses denoted average number of clauses, Mod. (LCG) represents average modularity of LCG graphs, and Mod. (VCG) represents average modularity of VCG graphs.

| Dataset | Easy | | | | Medium | | | | Hard | | | |
|---|---|---|---|---|---|---|---|---|---|---|---|---|
| | # Variables | # Clauses | Mod. (LCG) | Mod. (VCG) | # Variables | # Clauses | Mod. (LCG) | Mod. (VCG) | # Variables | # Clauses | Mod. (LCG) | Mod. (VCG) |
| $k$-Clique | 35.69 | 613.25 | 0.49 | 0.46 | 70.86 | 2298.03 | 0.49 | 0.48 | 114.49 | 5670.10 | 0.50 | 0.49 |
| $k$-Domset | 40.73 | 345.75 | 0.53 | 0.47 | 89.70 | 1708.06 | 0.51 | 0.49 | 137.32 | 4025.85 | 0.51 | 0.49 |
| $k$-Vercov | 46.33 | 498.06 | 0.52 | 0.48 | 108.19 | 2681.55 | 0.51 | 0.49 | 192.57 | 8409.32 | 0.51 | 0.50 |
| $k$-Color | 33.91 | 112.64 | 0.69 | 0.65 | 69.92 | 321.25 | 0.71 | 0.68 | 112.16 | 719.32 | 0.69 | 0.66 |
| $k$-Indset | 38.38 | 702.92 | 0.49 | 0.46 | 72.55 | 2388.22 | 0.49 | 0.48 | 113.12 | 5549.79 | 0.50 | 0.49 |
| Matching | 27.48 | 95.03 | 0.69 | 0.59 | 30.92 | 107.67 | 0.70 | 0.61 | 45.48 | 169.49 | 0.72 | 0.64 |
| Automorph | 56.76 | 943.54 | 0.51 | 0.47 | 82.74 | 1856.26 | 0.51 | 0.48 | 121.56 | 3612.56 | 0.51 | 0.49 |

Moreover, to evaluate the effectiveness of the learned representations on unseen SAT instances, we synthetically generate four more SAT datasets, including two random problems and two pseudo-industrial problems. Specifically, for random problems, we generate the SR dataset with the SR generator in NeuroSAT (Selsam et al., 2019), and the 3-SAT dataset with the 3-SAT generator in CNFGen (Lauria et al., 2017). For pseudo-industrial problems, we generate the CA dataset via the Community Attachment model (Giráldez-Cru & Levy, 2015), and the PS dataset by the Popularity-Similarity model (Giráldez-Cru & Levy, 2017). The generation process of the four datasets follows Li et al. (2023), where the dataset descriptions and statistics can also be found.

The ground truth of satisfiability and satisfying assignments are calculated by calling the state-of-the-art modern SAT solver CaDiCaL (Fleury & Heisinger, 2020), and the truth labels for unsat core variables are generated by invoking the proof checker DRAT-trim (Wetzler et al., 2014).

# B MORE DETAILS ON MODEL ARCHITECTURE

## B.1 INITIAL VERTEX FEATURES

As illustrated in the main paper, the input graphs primarily provide edge information instead of vertex features. Therefore, we should devise initial vertex features for the models. In this section, we introduce the definition of initial vertex features for the graph and SAT models.

**Graph Model Vertex Feature.** We begin by generating a normalized, learnable $d$-dimensional vector, which serves as the initial embedding shared across all vertices. For GDP datasets that do not require additional problem-specific information, such as Matching and Automorph, this initial embedding is directly used as the vertex feature for all vertices. In contrast, for GDP datasets where the parameter $k$ plays a critical role in defining the instance characteristics, such as $k$-Clique and $k$-Vercov, we first embed $k$ into a $d$-dimensional vector. The initial vertex embedding is then fused with the $k$ embedding through an MLP to generate the final initial vertex features.

**SAT Model Vertex Feature.** For the SAT model, we generate initial vertex features based on the type of SAT graph representation, whether it is a Literal-Clause Graph (LCG) or a Variable-Clause Graph (VCG). In the case of the LCG graph, we initialize a normalized, learnable $d$-dimensional vector for all literal nodes and a separate normalized, learnable $d$-dimensional vector for all clause nodes. Similarly, for the VCG graph, we generate a normalized, learnable $d$-dimensional vector for all variable nodes and another for all clause nodes.

## B.2 MORE BACKBONES

To demonstrate that the performance improvement brought about by our CORAL is consistent, and independent with specialized model architectures, we conduct experiments on more backbones.

**Graph Model Backbone.** For the graph model, we employ an additional mainstream network architecture for node embedding, GraphSAGE (Hamilton et al., 2017a), which is widely recognized for its ability to generate inductive representations of graph nodes by aggregating information from a node's local neighborhood. The update rule for the $i$-th layer of GraphSAGE is defined as follows:

$$\mathbf{n}_u^{(i)} = \text{AGG}\left(\text{ReLU}\left(\mathbf{Q}^{(i)}\mathbf{h}_v^{(i)} + \mathbf{q}^{(i)} \mid v \in N(u)\right)\right), \tag{7}$$

$$\mathbf{h}_u^{(i+1)} = \text{ReLU}\left(\mathbf{W}^{(i)}\,\text{CONCAT}\left(\mathbf{h}_u^{(i)}, \mathbf{n}_u^{(i)}\right)\right), \tag{8}$$

where $\mathbf{h}_u$ denotes the embedding for vertex $u$, $N(u)$ refers to the neighbors of vertex $u$, $\mathbf{Q}, \mathbf{q}, \mathbf{W}$ are trainable parameters, and AGG is the aggregation function. In our implementation, AGG is defined as the mean function, which computes the element-wise average of the neighbor embeddings.

**SAT Model Backbone.** For the SAT model, we incorporate a GCN architecture specifically tailored for SAT graphs as an additional backbone. The node updates at the $i$-th layer are defined as follows:

$$\mathbf{h}_l^{(i)} = \text{MLP}\left(\text{SUM}_{c \in \mathcal{N}(l)}\left(\text{MLP}\left(\mathbf{h}_c^{i-1}\right)\right), \mathbf{h}_l^{(i-1)}, \mathbf{h}_{\neg l}^{(i-1)}\right), \tag{9}$$

$$\mathbf{h}_c^{(i)} = \text{MLP}\left(\text{SUM}_{l \in \mathcal{N}(c)}\left(\text{MLP}\left(\mathbf{h}_l^{i-1}\right)\right), \mathbf{h}_c^{(i-1)}\right), \tag{10}$$

where $l$ and $c$ represent an arbitrary literal node and clause node, respectively. The aggregation of neighboring node information is performed using the summation operator (SUM), which serves as the aggregation function. The updates for both literal and clause nodes are computed using an MLP.

Furthermore, we extend the backbone to VCG graph modeling, where all literal nodes are replaced by variable nodes, and each literal and its negation are merged into a single variable node. The node updates at the $i$-th layer of the VGC-based GCN are formulated as:

$$\mathbf{h}_v^{(i)} = \text{MLP}\left(\text{SUM}_{c \in \mathcal{N}(v)}\left(\text{MLP}\left(\mathbf{h}_c^{i-1}\right)\right), \mathbf{h}_v^{(i-1)}\right), \tag{11}$$

$$\mathbf{h}_c^{(i)} = \text{MLP}\left(\text{SUM}_{v \in \mathcal{N}(c)}\left(\text{MLP}\left(\mathbf{h}_v^{i-1}\right)\right), \mathbf{h}_c^{(i-1)}\right), \tag{12}$$

where $v$ and $c$ represent an arbitrary variable node and clause node, respectively.

### B.3 CASE STUDY ON MODEL OUTUT

In this section, we illustrate the model outputs for specific GDP and corresponding SAT problems for better understanding.

In the context of GDP, the model's output is typically binary, represented as 0 or 1, at the instance level. For instance, in the case of the $k$-Clique problem, the input consists of a graph, and the output indicates whether the graph contains a clique of size $k$. Specifically, if a $k$-Clique is present, the output is 1; otherwise, it is 0.

Similarly, for the corresponding SAT problem, the output denotes the satisfiability of the formula. If the formula is satisfiable, the output is 1; if not, it is 0. The satisfiability result is directly linked to the solution of the original GDP problem. For example, a satisfiable formula indicates the existence of a $k$-Clique in the original graph.

However, the framework is not restricted to this specific task alone. By making appropriate modifications to the architecture of the output module, the models can be adapted to solve other related tasks, including both SAT-based and GDP-based tasks.

## C  LOSS FUNCTION FOR SAT-BASED TASKS

For the unsat core variable prediction task, we manually generate labels for the datasets, and adopt a binary cross-entropy loss on the label and the prediction.

For the satisfying assignment prediction task, we employ an unsupervised loss function as defined in Ozolins et al. (2022):

$$V_c(x) = 1 - \prod_{i \in c^+} (1 - x_i) \prod_{i \in c^-} x_i, \quad \mathcal{L}_\phi(x) = -\log\left(\prod_{c \in \phi} V_c(x)\right) = -\sum_{c \in \phi} \log\left(V_c(x)\right) \quad (13)$$

where $\phi$ refers to the CNF formula, $x$ is the predicted assignment consisting of binary values (0 or 1) for variables, $c$ denotes an arbitrary clause. The sets $c^+$ and $c^-$ comprise the variables present in clause $c$ in positive and negative forms, respectively. It is important to note that the loss function achieves its minimum value only when the predicted assignment $x$ corresponds to a satisfying assignment. Minimizing this loss can effectively aid in constructing a possible satisfying assignment.

# D  MORE EXPERIMENTAL RESULTS

## D.1  TRAINING PARAMETERS

For reproducibility, we present some important parameters used for training in Table 8. More details can be found in our source code, which will be released once the paper is accepted.

Table 8: Parameters used for training.

| Parameter | Value | Description |
|---|---|---|
| lr | 1e-04 | Learning rate. |
| lr_step_size | 50 | Learning rate step size. |
| lr_factor | 0.5 | Learning rate factor. |
| lr_patience | 10 | Learning rate patience. |
| clip_norm | 1.0 | Clipping norm. |
| weight_decay | 1e-08 | L2 regularzation weight. |
| sat_model_gnn_layer | 32 | Number of GNN layers in SAT model. |
| graph_model_gnn_layer | 12 | Number of GNN layers in graph model. |
| mlp_layer | 2 | Number of Linear layers in an MLP. |
| $\tau$ | 0.1 (easy) / 0.5 (medium) | Temperature scalar in the contrastive loss. |
| $\beta$ | 0.5∼1.0 | Weight of the decision loss during training. |

## D.2  COMPUTATIONAL COST

All training and inference tasks were conducted on a single NVIDIA H100 GPU with 80GB of memory.

The pre-training process for the SAT model and the graph models with CORAL totally takes approximately 40 hours, with convergence typically occurring around the 20th epoch. Each epoch requires roughly 2 hours. Following the pre-training phase, fine-tuning takes an additional 5 to 6 hours for each model to achieve optimal performance. In comparison, training the baseline SAT model takes about 45 hours, with convergence reached by the 30th epoch, and each epoch requiring approximately 1.5 hours. Notably, pre-training with CORAL demonstrates a faster convergence rate, leading to a shorter training time. Moreover, training the baseline graph model independently each requires around 15 hours, with convergence occurring around the 60th epoch, and each epoch taking between 12 to 18 minutes.

Overall, the computational cost of training with CORAL is comparable to that of the conventional training approach, with no significant increase in computational burden.

## D.3  MORE GENERALIZATION RESULTS

We show more results on generalization on the hard datasets in Table 9. Our approach to performance improvement is consistent across different model backbones. The SAT model with the GCN backbone exhibits minimal generalization capability across different problem difficulty levels. However, with other backbones, our approach consistently shows improved generalization performance compared to the baseline.

Table 9: Generalization performance across various model backbones on the hard datasets. The table presents the GDP-solving accuracy for the graph models and the satisfiability prediction accuracy for the SAT models. 'SAT Back.' refers to SAT model backbone, and 'Graph Back.' denotes graph model backbone. The terms 'Easy' and 'Medium' in parentheses indicate the difficulty level of the datasets used for training. The 'Overall' column represents the average accuracy across all datasets.

| SAT Backbone | Graph Backbone | Model | k-Clique | k-Domset | k-Vercov | k-Color | k-Indset | Matching | Automorph | Overall |
|---|---|---|---|---|---|---|---|---|---|---|
| LCG+GCN | GCN | Graph Model (Easy) | **0.545** | 0.500 | 0.500 | 0.546 | **0.505** | 0.664 | 0.631 | 0.556 |
| | | Graph Model+Contrast (Easy) | 0.525 | 0.500 | **0.539** | **0.557** | 0.499 | **0.686** | 0.631 | **0.562** |
| | | Graph Model (Medium) | 0.571 | 0.562 | 0.500 | 0.637 | 0.531 | 0.683 | 0.632 | 0.588 |
| | | Graph Model+Contrast (Medium) | **0.579** | **0.589** | **0.574** | **0.656** | **0.552** | **0.712** | **0.645** | **0.615** |
| | | SAT Model (Easy) | 0.500 | 0.500 | 0.500 | 0.459 | 0.500 | 0.539 | 0.500 | 0.500 |
| | | SAT Model+Contrast (Easy) | 0.500 | **0.592** | 0.500 | **0.500** | 0.500 | **0.591** | **0.513** | **0.528** |
| | | SAT Model (Medium) | 0.500 | 0.500 | 0.500 | 0.494 | 0.500 | 0.470 | 0.500 | 0.495 |
| | | SAT Model+Contrast (Medium) | 0.500 | 0.500 | 0.500 | **0.526** | 0.500 | **0.499** | 0.500 | **0.504** |
| VCG+GCN | GCN | Graph Model (Easy) | **0.545** | 0.500 | 0.500 | 0.546 | **0.505** | 0.664 | 0.631 | 0.556 |
| | | Graph Model+Contrast (Easy) | 0.531 | 0.500 | 0.500 | **0.554** | 0.496 | **0.684** | **0.634** | **0.557** |
| | | Graph Model (Medium) | 0.571 | 0.562 | 0.500 | 0.637 | 0.531 | 0.683 | 0.632 | 0.588 |
| | | Graph Model+Contrast (Medium) | **0.577** | **0.605** | **0.577** | **0.648** | **0.536** | **0.690** | **0.643** | **0.611** |
| | | SAT Model (Easy) | 0.500 | 0.500 | 0.500 | 0.500 | 0.500 | 0.500 | 0.500 | 0.500 |
| | | SAT Model+Contrast (Easy) | 0.500 | 0.500 | 0.500 | 0.500 | 0.500 | 0.500 | 0.500 | 0.500 |
| | | SAT Model (Medium) | 0.500 | 0.500 | 0.500 | 0.500 | 0.500 | 0.500 | 0.500 | 0.500 |
| | | SAT Model+Contrast (Medium) | 0.500 | 0.500 | 0.500 | **0.503** | 0.500 | 0.500 | 0.500 | 0.500 |
| LCG+NeuroSAT | GraphSAGE | Graph Model (Easy) | 0.509 | 0.503 | 0.481 | 0.508 | 0.505 | 0.578 | 0.557 | 0.520 |
| | | Graph Model+Contrast (Easy) | **0.529** | **0.599** | **0.559** | **0.602** | **0.585** | **0.679** | **0.621** | **0.596** |
| | | Graph Model (Medium) | 0.509 | 0.573 | 0.547 | 0.502 | 0.489 | 0.584 | 0.558 | 0.537 |
| | | Graph Model+Contrast (Medium) | **0.597** | **0.595** | **0.603** | **0.702** | **0.564** | **0.684** | **0.642** | **0.627** |
| | | SAT Model (Easy) | 0.475 | 0.505 | 0.500 | 0.588 | 0.473 | 0.995 | 0.729 | 0.609 |
| | | SAT Model+Contrast (Easy) | **0.596** | 0.505 | 0.500 | **0.615** | **0.587** | **0.996** | **0.821** | **0.660** |
| | | SAT Model (Medium) | 0.692 | 0.964 | 0.852 | 0.679 | 0.694 | 0.996 | 0.990 | 0.838 |
| | | SAT Model+Contrast (Medium) | **0.793** | **0.973** | **0.891** | **0.731** | **0.793** | 0.996 | **0.996** | **0.882** |

## D.4 MORE SAT-BASED TASK RESULTS.

We show more results on the satisfying assignment prediction task and the unsat core variable prediction task in Fig. 6. Our approach outperforms the baseline models with faster convergence and higher final accuracy.

## D.5 FURTHER STUDY ON GNN BACKBONE

To further assess the efficacy of CORAL, we implement two more advanced GNN backbones, PGN (Veličković et al., 2020) and GraphGPS (Rampášek et al., 2022), for our graph models. All related experiments presented in the main paper are conducted. The results are summarized in Table 10 and Table 11. PGN achieves performance comparable to GraphSAGE, while GraphGPS significantly outperforms the other backbones. Notably, CORAL consistently improves performance on accuracy and generalization ability across both backbones, thereby demonstrating its effectiveness regardless of the underlying GNN architecture.

Table 10: Experimental results across various model backbones. The table presents the GDP-solving accuracy for the graph models and the satisfiability prediction accuracy for the SAT models. 'SAT Back.' refers to SAT model backbone, and 'Graph Back.' denotes graph model backbone.

| SAT Back. | Graph Back. | Difficulty | Model | k-Clique | k-Domset | k-Vercov | k-Color | k-Indset | Matching | Automorph | Overall |
|---|---|---|---|---|---|---|---|---|---|---|---|
| LCG+NeuroSAT | PGN | Easy | Graph Model | 0.762 | 0.584 | 0.664 | 0.916 | 0.679 | 0.687 | 0.617 | 0.701 |
| | | | Graph Model+Contrast | **0.773** | **0.619** | **0.697** | **0.937** | **0.716** | **0.703** | 0.617 | **0.723** |
| | | Medium | Graph Model | **0.724** | 0.628 | 0.647 | 0.830 | **0.681** | 0.588 | 0.504 | 0.657 |
| | | | Graph Model+Contrast | 0.720 | **0.633** | **0.660** | **0.864** | 0.672 | **0.708** | **0.633** | **0.699** |
| | | Easy | SAT Model | 0.959 | 0.991 | 0.998 | 0.974 | 0.954 | 0.995 | 0.999 | 0.981 |
| | | | SAT Model+Contrast | **0.989** | **0.996** | 0.998 | **0.988** | **0.991** | **0.999** | 0.999 | **0.994** |
| | | Medium | SAT Model | 0.876 | 0.987 | 0.991 | 0.817 | 0.887 | 0.997 | 0.988 | 0.935 |
| | | | SAT Model+Contrast | **0.905** | **0.990** | **0.995** | **0.941** | **0.914** | 0.999 | **0.997** | **0.963** |
| LCG+NeuroSAT | GraphGPS | Easy | Graph Model | 0.824 | 0.772 | 0.855 | 0.899 | 0.764 | 0.694 | **0.674** | 0.783 |
| | | | Graph Model+Contrast | **0.839** | **0.774** | **0.885** | **0.906** | **0.784** | **0.763** | 0.664 | **0.802** |
| | | Medium | Graph Model | 0.707 | 0.625 | 0.658 | 0.849 | 0.618 | **0.694** | 0.626 | 0.682 |
| | | | Graph Model+Contrast | **0.717** | **0.729** | **0.818** | **0.856** | **0.730** | 0.572 | **0.632** | **0.722** |
| | | Easy | SAT Model | 0.959 | 0.991 | 0.998 | 0.974 | 0.954 | 0.995 | 0.999 | 0.981 |
| | | | SAT Model+Contrast | **0.986** | **0.996** | **0.999** | **0.985** | **0.987** | **0.998** | 0.999 | **0.993** |
| | | Medium | SAT Model | 0.876 | 0.987 | 0.991 | 0.817 | 0.887 | 0.997 | 0.988 | 0.935 |
| | | | SAT Model+Contrast | **0.914** | **0.990** | **0.996** | **0.939** | **0.922** | 0.997 | **0.996** | **0.965** |

Table 11: Generalization performance across various model backbones on the hard datasets. The table presents the GDP-solving accuracy for the graph models and the satisfiability prediction accuracy for the SAT models. 'SAT Back.' refers to SAT model backbone, and 'Graph Back.' denotes graph model backbone. The terms 'Easy' and 'Medium' in parentheses indicate the difficulty level of the datasets used for training. The 'Overall' column represents the average accuracy across all datasets.

| SAT Backbone | Graph Backbone | Model | k-Clique | k-Domset | k-Vercov | k-Color | k-Indset | Matching | Automorph | Overall |
|---|---|---|---|---|---|---|---|---|---|---|
| LCG+NeuroSAT | PGN | Graph Model (Easy) | 0.542 | 0.593 | 0.595 | 0.631 | 0.549 | 0.663 | 0.603 | 0.597 |
| | | Graph Model+Contrast (Easy) | **0.546** | **0.598** | **0.599** | **0.633** | **0.551** | **0.667** | **0.610** | **0.601** |
| | | Graph Model (Medium) | 0.604 | 0.586 | 0.597 | 0.691 | 0.559 | 0.671 | **0.635** | 0.620 |
| | | Graph Model+Contrast (Medium) | **0.612** | **0.589** | **0.607** | **0.697** | **0.581** | **0.675** | 0.633 | **0.628** |
| | | SAT Model (Easy) | 0.475 | 0.505 | 0.500 | 0.588 | 0.473 | **0.995** | 0.729 | 0.609 |
| | | SAT Model+Contrast (Easy) | **0.597** | **0.507** | 0.500 | **0.614** | **0.596** | 0.979 | **0.772** | **0.652** |
| | | SAT Model (Medium) | 0.692 | 0.964 | 0.852 | 0.679 | 0.694 | 0.996 | 0.990 | 0.838 |
| | | SAT Model+Contrast (Medium) | **0.787** | **0.974** | **0.900** | **0.736** | **0.796** | **0.998** | **0.993** | **0.883** |
| LCG+NeuroSAT | GraphGPS | Graph Model (Easy) | **0.596** | 0.500 | 0.499 | 0.500 | 0.535 | **0.680** | 0.576 | 0.555 |
| | | Graph Model+Contrast (Easy) | 0.593 | **0.507** | **0.609** | **0.596** | 0.535 | 0.589 | **0.595** | **0.575** |
| | | Graph Model (Medium) | 0.632 | 0.552 | 0.568 | 0.683 | 0.630 | **0.639** | 0.583 | 0.612 |
| | | Graph Model+Contrast (Medium) | **0.638** | **0.608** | **0.779** | **0.689** | **0.657** | 0.601 | **0.614** | **0.655** |
| | | SAT Model (Easy) | 0.475 | 0.505 | 0.500 | 0.588 | 0.473 | **0.995** | 0.729 | 0.609 |
| | | SAT Model+Contrast (Easy) | **0.505** | **0.506** | **0.504** | **0.596** | **0.503** | 0.993 | **0.762** | **0.624** |
| | | SAT Model (Medium) | 0.692 | 0.964 | 0.852 | 0.679 | 0.694 | **0.996** | 0.990 | 0.838 |
| | | SAT Model+Contrast (Medium) | **0.760** | **0.969** | **0.961** | **0.738** | **0.760** | 0.994 | **0.993** | **0.882** |

## D.6 FURTHER STUDY ON CONTRASTIVE LOSS

We revise the negative sampling strategy within our contrastive learning framework to mitigate the issue of false negative samples. Specifically, within each training batch, unsatisfiable instances are selected as negative samples for satisfiable instances, and conversely, satisfiable instances are chosen as negative samples for unsatisfiable instances. This adjustment ensures that false negative samples are avoided. Consequently, we modify the contrastive loss function to reflect this change and proceed with the training of the models. The results, as shown in Table 12, demonstrate that the models trained with the revised contrastive loss exhibit performance comparable to that of those trained with the original loss. We also plot the contrastive loss curves for several GDPs during the original training process in Fig. 4, all of which exhibit smooth trajectories. These results suggest that the influence of false negative samples on model performance is minimal.

Table 12: Experimental results on the modified and original contrastive loss function. The table presents the GDP-solving accuracy for the graph models and the satisfiability prediction accuracy for the SAT models. 'Graph/SAT Model+Contrast+Modified Loss' denotes training with the modified contrastive loss. 'SAT Back.' refers to SAT model backbone, and 'Graph Back.' denotes graph model backbone.

| SAT Back. | Graph Back. | Difficulty | Model | k-Clique | k-Domset | k-Vercov | k-Color | k-Indset | Matching | Automorph | Overall |
|---|---|---|---|---|---|---|---|---|---|---|---|
| LCG+NeuroSAT | GCN | Easy | Graph Model+Contrast+Modified Loss | 0.771 | 0.579 | 0.615 | 0.887 | 0.642 | 0.715 | 0.644 | 0.693 |
| | | | Graph Model+Contrast | **0.793** | **0.620** | **0.673** | **0.902** | **0.675** | **0.717** | **0.654** | **0.719** |
| | | Medium | Graph Model+Contrast+Modified Loss | 0.707 | 0.630 | 0.612 | 0.798 | 0.589 | 0.724 | 0.637 | 0.671 |
| | | | Graph Model+Contrast | **0.713** | **0.646** | **0.633** | **0.822** | **0.640** | **0.728** | **0.657** | **0.691** |
| | | Easy | SAT Model+Contrast+Modified Loss | 0.983 | **0.996** | 0.999 | 0.985 | 0.981 | 0.999 | 0.999 | 0.992 |
| | | | SAT Model+Contrast | **0.989** | 0.996 | 0.999 | **0.988** | **0.989** | 0.999 | 0.999 | **0.994** |
| | | Medium | SAT Model+Contrast+Modified Loss | 0.907 | 0.991 | 0.995 | 0.923 | 0.917 | 0.999 | 0.999 | 0.960 |
| | | | SAT Model+Contrast | **0.923** | 0.991 | 0.996 | **0.946** | **0.930** | 0.999 | 0.999 | **0.969** |

## D.7 FURTHER STUDY ON GRAPH MODEL GENERALIZATION

In the main paper, we demonstrate the generalization capabilities of our models across different difficulty levels, as well as the ability to generalize to unseen domains of the SAT model. In this section, we conduct a more comprehensive evaluation of the generalization ability of the graph models through two additional experimental settings.

### D.7.1 GENERALIZATION ON RELATED TASK

We select the $k$-Clique and $k$-Vertex Cover problems as the original problem domains and design two related tasks to assess generalization. For the $k$-Clique problem, we adapt the pre-trained model to predict the maximum clique size in the input graph. For the $k$-Vertex Cover problem, the model

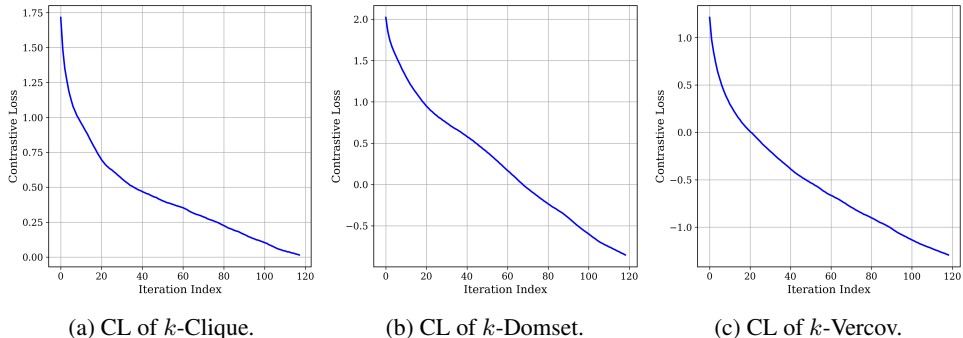

(a) CL of $k$-Clique.  (b) CL of $k$-Domset.  (c) CL of $k$-Vercov.

Figure 4: Contrastive loss w.r.t. training iterations across various datasets. CL denotes the contrastive loss of the training process.

is tasked with predicting the minimum number of vertices required to cover the edges of the input graph. The pre-trained model is fine-tuned using a subset of $\frac{1}{8}$ of the training data. For comparing the performance of the fine-tuned models with models trained from scratch with full training data, we employ the mean relative error (MRE):

$$\text{Mean Relative Error} = \frac{1}{N} \sum_{i=1}^{N} |\frac{y_i - \hat{y}_i}{y_i}| \tag{14}$$

where $y_i$ refers to the ground truth, $\hat{y}_i$ refers to the predicted value, $N$ refers to the sample size. Figure 5 illustrates that the pre-trained model achieves faster convergence and superior final performance, underscoring its enhanced generalization ability.

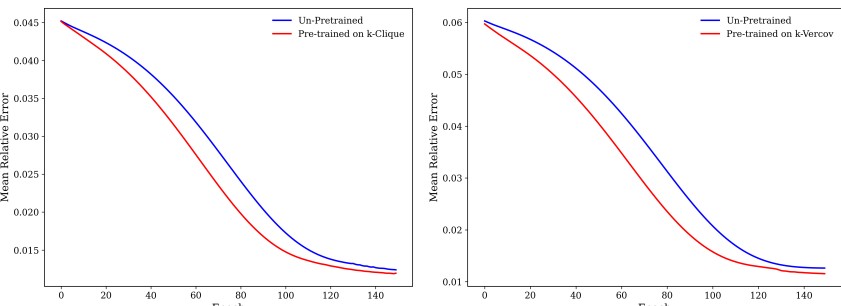

(a) MRE of Max Clique Size Prediction.  (b) MRE of Min Vertex Number Prediction.

Figure 5: Mean Relative error w.r.t. epoch on related graph tasks, including maximum clique size prediction and minimum vertex number prediction for edge cover. MRE denotes the mean relative error evaluated on test set data.

### D.7.2 GENERALIZATION ON LARGE-SCALE DATA

To further assess the generalization ability of our graph models, we generate large-scale instances for each GDP, with instance sizes ranging from 7 to 20 times larger than those used during pre-training. We then fine-tune the pre-trained models on this large-scale data, using a subset comprising $\frac{1}{8}$ of the training data. We compare the performance of the fine-tuned models with those trained from scratch with full training data, and the results are presented in Table 13, indicating that models pre-trained on smaller instances using CORAL can generalize effectively to larger instances through fine-tuning.

### D.8 FURTHER STUDY ON MODEL SENSITIVITY

The solution to GDP is known to be sensitive to graph structures. Therefore, we aim to evaluate the sensitivity of our model to perturbations in graph structure. To do so, we generate modified instances

Table 13: Experimental results across two graph models under different training methods. 'Graph Model (fully-trained)' refers to the graph model that was trained from scratch with full training data. 'Graph Model+Contrast (fine-tuned)' refers to the fine-tuned graph model after pre-training by CORAL on small datasets.

| Model | k-Clique | k-Domset | k-Vercov | k-Color | k-Indset | Matching | Automorph | Overall |
|---|---|---|---|---|---|---|---|---|
| Graph Model (fully-trained) | 0.673 | 0.667 | 0.654 | 0.791 | 0.591 | 0.724 | 0.654 | 0.679 |
| Graph Model+Contrast (fine-tuned) | **0.679** | **0.670** | **0.666** | **0.794** | **0.615** | **0.726** | **0.657** | **0.687** |

by adding or removing edges from the original graphs until either the satisfiability status reverses or the number of modified edges reaches $\frac{1}{10}$ of the original edge count. These generated instances are structurally similar to the original graphs but exhibit a reversed satisfiability status. We then assess the performance of both the graph models and the SAT model on these perturbed instances. The results, presented in Table 14, reveal that the SAT model is sensitive to changes in graph structure, and it continues to perform well. Additionally, the graph models significantly outperform the baseline models, as they are more closely aligned with the SAT model and demonstrate enhanced sensitivity to structural changes.

Table 14: Experimental results on perturbed instances. The table presents the GDP-solving accuracy for the graph models and the satisfiability prediction accuracy for the SAT models on perturbed instances. 'SAT Back.' refers to SAT model backbone, and 'Graph Back.' denotes graph model backbone.

| SAT Back. | Graph Back. | Model | k-Clique | k-Domset | k-Vercov | k-Color | k-Indset | Matching | Automorph | Overall |
|---|---|---|---|---|---|---|---|---|---|---|
| LCG+NeuroSAT | GCN | Graph Model | 0.652 | 0.511 | 0.534 | 0.614 | 0.536 | 0.588 | 0.377 | 0.545 |
| | | Graph Model+Contrast | **0.678** | **0.547** | **0.618** | **0.719** | **0.664** | **0.656** | **0.421** | **0.615** |
| | | SAT Model | 0.976 | 0.923 | 0.982 | 0.933 | 0.971 | 0.854 | 0.923 | 0.937 |
| | | SAT Model+Contrast | **0.983** | **0.940** | **0.997** | **0.939** | **0.984** | **0.861** | **0.939** | **0.949** |

## D.9 ABLATION STUDY ON CROSS-DOMAIN INFORMATION TRANSFER

A central component of our framework is the facilitation of information transfer across different problem domains. To evaluate the effectiveness of this mechanism, we conduct an ablation study by disabling the cross-domain information transfer. Specifically, we train each graph model independently with its own SAT model, without leveraging cross-domain information. We then compare this ablated approach with our original method, as shown in Table 15. The results indicate that the ablated approach yields inferior performance, thereby highlighting the importance and effectiveness of the cross-domain information transfer in enhancing the model's performance.

Table 15: Ablation study on cross-domain information transfer. The table presents the GDP-solving accuracy for the graph models and the satisfiability prediction accuracy for the SAT models. 'Graph/SAT Model+Single Domain' refers to the ablated method by disabling cross-domain information transfer. 'SAT Back.' refers to SAT model backbone, and 'Graph Back.' denotes graph model backbone.

| SAT Back. | Graph Back. | Difficulty | Model | k-Clique | k-Domset | k-Vercov | k-Color | k-Indset | Matching | Automorph | Overall |
|---|---|---|---|---|---|---|---|---|---|---|---|
| LCG+NeuroSAT | GCN | Easy | Graph Model+Single Domain | 0.784 | 0.617 | 0.671 | 0.899 | 0.656 | 0.715 | 0.653 | 0.714 |
| | | | Graph Model+Contrast | **0.793** | **0.620** | **0.673** | **0.902** | **0.675** | **0.717** | **0.654** | **0.719** |
| | | Medium | Graph Model+Single Domain | 0.709 | 0.640 | 0.629 | 0.810 | 0.599 | 0.725 | 0.643 | 0.679 |
| | | | Graph Model+Contrast | **0.713** | **0.646** | **0.633** | **0.822** | **0.640** | **0.728** | **0.657** | **0.691** |
| | | Easy | SAT Model+Single Domain | 0.987 | 0.996 | 0.999 | 0.988 | 0.988 | 0.999 | 0.999 | 0.994 |
| | | | SAT Model+Contrast | **0.989** | 0.996 | 0.999 | 0.988 | **0.989** | 0.999 | 0.999 | 0.994 |
| | | Medium | SAT Model+Single Domain | 0.906 | 0.99 | 0.994 | 0.945 | 0.884 | 0.999 | 0.994 | 0.959 |
| | | | SAT Model+Contrast | **0.923** | **0.991** | **0.996** | **0.946** | **0.930** | 0.999 | **0.999** | **0.969** |

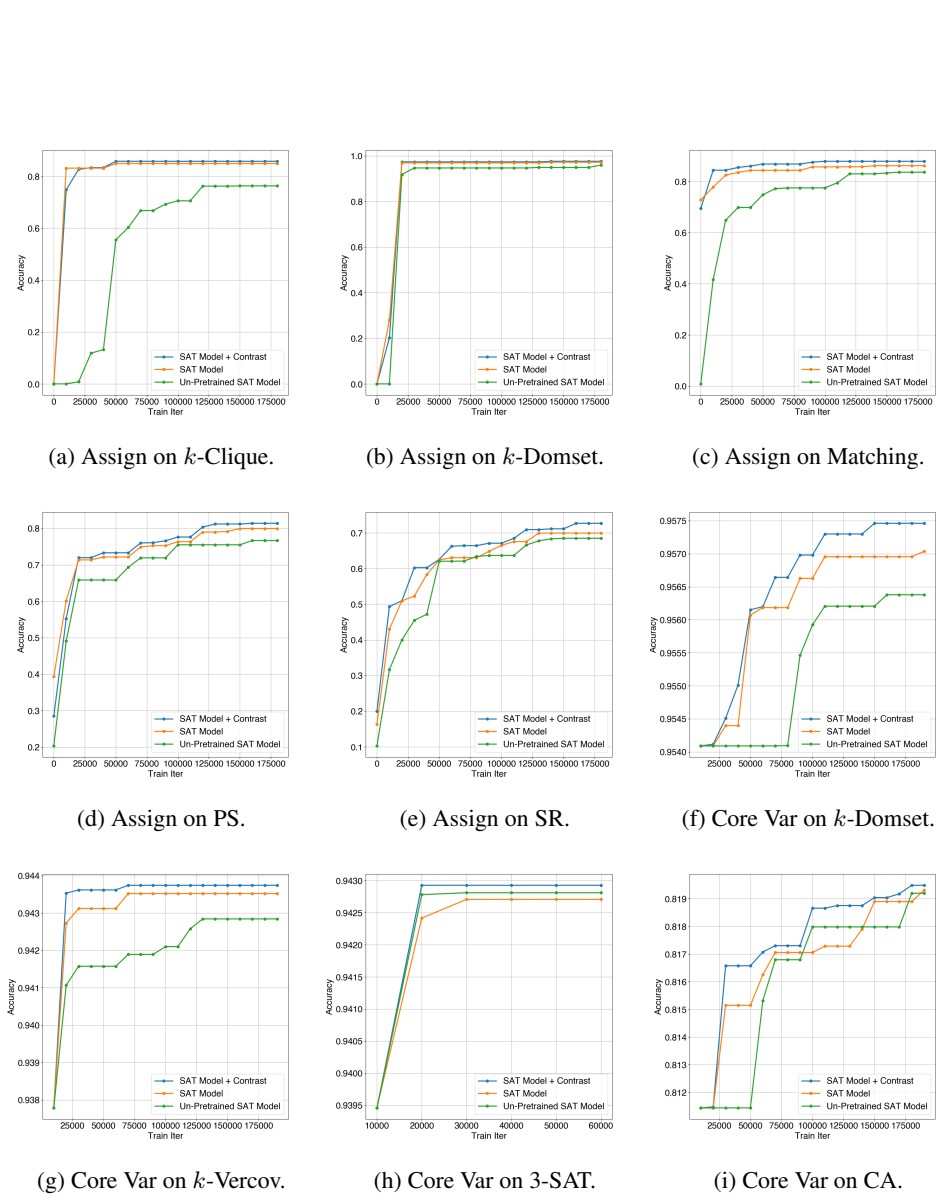

Figure 6: Model performance w.r.t. training iterations on SAT-based tasks across various datasets. Assign denotes the satisfying assignment prediction task, and Core Var denotes the unsat core variable prediction task.

