# OpenReview forum: "Learning General Representations Across Graph Combinatorial Optimization Problems"
_ICLR.cc/2025/Conference — Submitted to ICLR 2025_

### Official Review · Reviewer_MBJq · 2024-10-16

**Soundness:** 3
**Presentation:** 3
**Contribution:** 3
**Rating:** 3
**Confidence:** 4

**Summary:**

The paper introduces a novel paradigm called CORAL (Combinatorial Optimization Representation Alignment and Learning) for learning general representations across different combinatorial optimization (CO) problems. The main idea is to treat each graph decision problem (GDP) as a separate modality and use the Boolean satisfiability (SAT) problem as an intermediary to unify these problem types. The proposed CORAL framework employs cross-modal contrastive learning to align different GDPs through SAT representations, thereby enhancing the quality and generalizability of learned representations.

**Strengths:**

1. **Novel Unified Framework**: CORAL is the first to unify representation learning across different combinatorial optimization problems, which is innovative and addresses a significant gap in current methods.

2. **Effective Use of SAT as an Intermediary**: Leveraging SAT as a common modality effectively bridges the differences between various graph decision problems, promoting knowledge transfer and alignment.

3. **Enhanced Generalizability**: The cross-modal contrastive learning approach significantly improves the quality and generalizability of learned representations, as demonstrated by superior performance in both task-specific and generalization experiments.

4. **Comprehensive Experiments**: The paper provides extensive experimental validation across multiple problem types and tasks, highlighting the robustness and scalability of the proposed approach.

**Weaknesses:**

1. **Complexity of the Framework**: The introduction of SAT as an intermediary and cross-modal contrastive learning adds substantial complexity, potentially making the model challenging to implement and computationally expensive.

2. **Limited Comparison with Alternative Methods**: The paper lacks a comprehensive comparison with other state-of-the-art multi-task learning approaches or GNN-based optimization frameworks, limiting the understanding of CORAL's relative strengths.

3. **Scalability Concerns**: The approach may face scalability issues when dealing with larger, real-world problem instances due to the overhead of transforming GDPs into SAT representations and training multiple models.

4. **Insufficient Analysis of Computational Cost**: There is no detailed analysis of the training time, memory requirements, or computational resources needed for CORAL, which could be a barrier to real-world adoption.

5. **Lack of Ablation Studies**: The paper does not include an ablation study to evaluate the contribution of each component (e.g., SAT transformation, contrastive learning) to the overall performance, making it difficult to determine their individual impact.

**Questions:**

See above.

---

> ### Author Response · Authors · 2024-11-19
>
> We would like to express our sincere gratitude for thoroughly evaluating our paper and providing detailed and constructive feedback. We are genuinely committed to addressing your concerns and respond to your specific comments below.
>
> > **W1&W4: Complexity of the framework and insufficient analysis of computational cost.**
>
> Thank you for your feedback. **We have included detailed information on the training time and memory requirements in the revised [PDF](https://openreview.net/pdf?id=elmTU101oS) (in Appendix D.2).** The computational cost of training CORAL is comparable to seperately training graph and SAT models.
>
> From a conceptual standpoint, the design of our framework is clear and easy to implement. The reason we use separate models for each domain, particularly for GDPs, is that these problems often have strong dependencies on specific properties of the graphs. It is challenging to design a single, universal model that effectively captures all the necessary features across different problem domains. In order to create a model that can handle all types of problems, one would need to design domain-specific features for each problem, which requires significant expert knowledge and effort.
>
> To resolve this challenge, our framework treats the graph problem instances as a mapping from graph to decision, allowing the neural network to automatically learn the relevant properties of the graphs needed to solve the problem. This approach eliminates the need for manually designing features for each specific problem. Thus, while each model is domain-specific, the framework unifies the problem-solving process for all domains.
>
>
>
> > **W2: Limited comparison with alternative methods.**
>
> Thank you for your feedback. To the best of our knowledge, our work is the first to propose learning general representations across GDPs. During our literature review, we did not identify existing frameworks that could directly be adopted for comparison in the context of learning representations across GDPs. This uniqueness makes it challenging to compare our approach with prior works.
>
> In our experiments, we unified the backbones used across our framework and the baseline models to ensure a fair evaluation. This allows us to isolate and verify the effectiveness of our framework, including its ability to improve accuracy and generalization compared to task-specific models.
>
> **To further strengthen the evaluation and assist in better understanding the performance of our method, we have included additional experiments in the revised [PDF](https://openreview.net/pdf?id=elmTU101oS) (in Appendix D.5, D.6, D.7, D.8, D.9).**
>
>
>
> > **W3: Scalability concerns with larger, real-world problem instances.**
>
> Thank you for your feedback. To address the scalability concerns, **we have added a new experiment in the revised [PDF](https://openreview.net/pdf?id=elmTU101oS). This experiment shows that models pre-trained on small, easy-to-handle instances using CORAL can be effectively generalized on large instances through fine-tuning, without the need for transformations of GDPs into SAT (in Appendix D.7.2),** proving that our framework eliminates the need for transformations of GDPs into SAT representations and the training of multiple models for larger instances. The results demonstrate that our framework is scalable and can handle large-scale instances efficiently.
>
>
> > **W5: Lack of ablation studies.**
>
> Thank you for your comment. We would like to clarify that the SAT transformation and contrastive learning are inherently interlinked in our framework. As such, it is not feasible to conduct an ablation study where these components are tested independently.
>
> However, **we do have conduct a new ablation study to validate the effectiveness of cross-domain information transfer in our framework and add it in the revised [PDF](https://openreview.net/pdf?id=elmTU101oS) (in Appendix D.9).**
>
>
> We hope this response could help address your concerns and wish to receive your further feedback soon.

---

> ### Author Response · Authors · 2024-11-25
>
> Dear Reviewer MBJq,
>
> We extend our sincere appreciation for your time and the valuable insights you've provided. We are eager to ensure that our rebuttal adequately addresses all your concerns and are looking forward to engaging in a further discussion with you. Should any lingering questions persist or if further clarifications are needed, please don't hesitate to reach out.
>
> Best regards,
>
> The Authors

---

> ### Comment · Area_Chair_nLqp · 2024-11-25
> **ICLR Public Discussion Phase Ending Soon**
>
> Dear Reviewer,
>
> This is a kind reminder that the discussion phase will be ending soon on November 26th. Please read the author responses and engage in a constructive discussion with the authors.
>
> Thank you for your time and cooperation.
>
> Best,
>
> Area Chair

---

> ### Author Response · Authors · 2024-11-27
>
> Dear Reviewer MBJq,
>
> Thank you again for your valuable feedback on our submission. We appreciate the time and effort you’ve taken to evaluate our work.
>
> We understand that reviewing can be a time-consuming process, and we want to kindly follow up regarding the rebuttal we submitted a week ago. If you have any further questions or need clarification from us, we would be happy to provide additional information. However, if there are no further concerns, we would kindly request that you reconsider the rating in light of the clarifications we’ve provided.
>
> Thank you for your attention, and we look forward to hearing from you soon.
>
> Best regards,
>
> The Authors

---

### Official Review · Reviewer_95cx · 2024-10-28

**Soundness:** 3
**Presentation:** 3
**Contribution:** 3
**Rating:** 6
**Confidence:** 3

**Summary:**

This paper introduces CORAL, a novel learning paradigm that enhances generalizability and representation quality across various graph combinatorial optimization problems by unifying them as SAT problems through cross-modal contrastive learning.

**Strengths:**

1. This paper proposes CORAL, a new framework for learning universal representations of multiple CO problems.

2. This paper introduces SAT as an intermediate unified mode to bridge different CO problems and effectively learn the shared features.

3. Extensive experiments have been conducted on multiple problems and datasets to validate the effectiveness of the proposed method.

**Weaknesses:**

Please refer to questions.

**Questions:**

1. The goal of achieving high generalization through multiple losses[1] or collaborative representations of multiple models[2] seems to be a method that has been discussed multiple times outside of the GDP field. What is the core difference between the method proposed in this paper and those methods?

2. Why using the contrastive loss to align the representation of the GDP and SAT modalities is a way to get a higher-level, abstract representation?

3.  Are there any other comparable works in the experiment? There seems to be a lack of comparison with other similar types of work in the experiment.

[1] Automated Self-Supervised Learning for Graphs. Arxiv 2106.05470

[2] Decoupling Weighing and Selecting for Integrating Multiple Graph Pre-training Tasks. Arxiv 2403.01400


---

I work in graph pre-training, but I am not familiar with GDP problems. If authors can address my concerns, I will reconsider my score.

---

> ### Author Response · Authors · 2024-11-19
>
> We would like to express our sincere gratitude for thoroughly evaluating our paper and providing insightful and valuable feedback. We are genuinely committed to addressing your concerns and respond to your specific comments below.
>
> > **Q1: What is the core difference between the method proposed in this paper and graph pre-training methods?**
>
> Thank you for your feedback. Our approach also contains pre-training on graphs. However, compared to typical graph pre-training methods, our approach does not focus on task design or the structure of the graph itself, but instead learns on a rich external representation (SAT) and relations among different tasks, thus unifying the representations learned across multiple tasks and forming higher-level representations that encapsulate the shared problem features of diverse GDP tasks.
>
> To our understanding, the primary goal of graph pre-training is to extract informative knowledge from large-scale, unlabeled data using carefully designed pre-training (self-supervised) tasks and reduce the dependency on labeled data. For example, in graph pre-training for a GDP task, e.g., the k-clique problem, one might select self-supervised pre-training task combinations that may best align with this task, possibly including a pre-training task where the model predicts the size of the largest clique in a graph.
>
> In contrast, the challenge with GDPs lies in the difficulty of encoding the task type itself (since it is more related to logical reasoning and each task has unique characteristics) and the weak supervision signals available. For instance, GDP instances only provide instance-level labels (e.g., a binary 0 or 1 for a graph), which is insufficient for graph models to effectively learn task-specific properties.
>
> Our method addresses this issue by transforming all graph instances into unified SAT representations utilizing combinatorial optimization problem characteristics. The SAT representation explicitly encodes the logic of the problems, providing a rich source of information that allows the graph models to align. This alignment enables the graph models to better understand the task without requiring manually designed features or pre-training tasks, enabling the models to learn task-specific properties autonomously in a unified process. Also, aligning models for all tasks with the same SAT model enables the models to find and capture the relations among different tasks (similar to CLIP[1]) and unify the learned representations on different tasks, thereby forming higher-level representations that encapsulate the shared problem features.
>
> [1] Radford A, Kim J W, Hallacy C, et al. Learning transferable visual models from natural language supervision[C]//International conference on machine learning. PMLR, 2021: 8748-8763.
>
> > **Q2: Why using the contrastive loss to align the representation of the GDP and SAT modalities is a way to get a higher-level, abstract representation?**
>
> Thank you for your insightful question. The use of contrastive loss to align the representations of GDP and SAT modalities plays a central role in achieving a higher-level, abstract representation.
>
> All GDP problems can be transformed into SAT representations, which reflects the fact that they share a common underlying structure. This structure can potentially be leveraged to create a unified and more efficient representation of these problems. Through contrastive loss, the representations of different GDP problems are aligned to their SAT counterparts, and we can effectively map all problem types into a shared latent space, which captures the essential commonalities between different GDP problem types and encapsulates the higher-level, abstract representation that unifies the various GDP problems.
>
>
> > **Q3: Are there any other comparable works in the experiment?**
>
> Thank you for your feedback. To the best of our knowledge, our work is the first to propose learning general representations across GDPs. During our literature review, we did not identify existing frameworks that could directly be adopted for comparison in the context of learning representations across GDPs. This uniqueness makes it challenging to compare our approach with prior works.
>
> In our experiments, we unified the backbones used across our framework and the baseline models to ensure a fair evaluation. This allows us to isolate and verify the effectiveness of our framework, including its ability to improve accuracy and generalization compared to task-specific models.
>
> **To further strengthen the evaluation and assist in better understanding the performance of our method, we have included additional experiments in the revised [PDF](https://openreview.net/pdf?id=elmTU101oS) (in Appendix D.5, D.6, D.7, D.8, D.9).**
>
>
> We hope this response could help address your concerns and wish to receive your further feedback soon.

---

> > ### Comment · Reviewer_95cx · 2024-11-22
> > **Thanks for your response**
> >
> > I definitely will consider to increase my rating after the discussion phase.

---

> > > ### Author Response · Authors · 2024-11-22
> > >
> > > Dear Reviewer 95cx,
> > >
> > > We extend our gratitude for your generous support and for helping us improve the paper. We appreciate your valuable suggestions.
> > >
> > > We are open to any further questions or concerns you may have. If anything remains unclear or if you have additional feedback, please do not hesitate to let us know.
> > >
> > > Best regards,
> > >
> > > The Authors

---

### Official Review · Reviewer_qz26 · 2024-11-02

**Soundness:** 3
**Presentation:** 3
**Contribution:** 3
**Rating:** 6
**Confidence:** 3

**Summary:**

This paper focuses on solving graph deision problems (GDPs) with graph neural networks by modeling each GDP type with task-specific graph modules. The correponding SAT bipartite graphs of each GDP are utlized as an intermediate modal of data that contrastively optmizes the graph representation modules. Experimental results validate the effectiveness of the proposed graph GDP model in generating representations and solving GDP probelems.

**Strengths:**

+ A pioneering effort that propose to process GDP problems with GNN frameworks.
+ Combining GDP problems with corresponding SAT graphs is a novel and well motivated idea.
+ The method is well presented and easy to follow.

**Weaknesses:**

- The proposed model is introduced as a general solution to combinatorial optimization problems, yet it is limited to solving GDP problems. Although CO problems can be transformed GDPs in the sense of complexity, solving GDPs is not necessarily equivalent to practical solutions to correponding CO problems.
- It seems that the general SAT model significantly outperforms specific graph models, raising concerns about the expressiveness of the graph model.
- It would be more illustrative to have case studies on model outputs for specific GDP and correponding SAT problems.

**Questions:**

Please refer to the weaknesses.

---

> ### Author Response · Authors · 2024-11-19
>
> We would like to express our sincere gratitude for thoroughly evaluating our paper and providing valuable and constructive feedback. We are genuinely committed to addressing your concerns and respond to your specific comments below.
>
> > **W1: The model is limited to solving GDP problems.**
>
> Thanks for your feedback. The primary goal of our framework is to facilitate better representation learning for graph instances, which can then be applied to various downstream tasks. While the current focus of our work is on solving GDPs, the framework is not restricted to this specific task alone. By making appropriate modifications to the architecture of the output module, our approach can be adapted to solve other related tasks that are more directly aligned with practical CO problem-solving.
>
> **To address your concern, we have included additional experiments in the revised [PDF](https://openreview.net/pdf?id=elmTU101oS) where we evaluate our framework on two tasks that are related to GDPs but are more closely aligned with practical solutions to CO problems (in Appendix D.7.1).** These additional tasks demonstrate the framework’s potential to generalize beyond pure GDP problem-solving.
>
>
> > **W2: General SAT model significantly outperforms specific graph models.**
>
> Thank you for your insightful comment. The SAT model explicitly encodes the logical structure of the problem into the input representation, which provides direct features for learning. In contrast, the graph model’s input is limited to the raw graph instances without explicit features representing the problem itself. This is due to the fact that GDP problems are closely tied to specific graph properties, and designing features that are universally effective across all problems is highly challenging and requires extensive expert knowledge. Instead, we rely on the graph model to autonomously learn the graph characteristics needed to solve the problem, which inherently makes the problem-solving process more complex compared to the SAT model.
>
> Despite the higher difficulty for the graph models, our experimental results validate the effectiveness of our framework. The consistent performance improvements across different GDPs demonstrate that the graph models can effectively learn the necessary properties for solving GDP problems through our framework.
>
> **To address your concern and further evaluate the expressiveness of the graph model, we have conducted additional experiments using two more powerful GNN backbones[1][2] (in Appendix D.5).** The performance improvement is consistent regardless of the backbone model.
>
> [1] Veličković P, Buesing L, Overlan M, et al. Pointer graph networks[J]. Advances in Neural Information Processing Systems, 2020, 33: 2232-2244.
>
> [2] Rampášek L, Galkin M, Dwivedi V P, et al. Recipe for a general, powerful, scalable graph transformer[J]. Advances in Neural Information Processing Systems, 2022, 35: 14501-14515.
>
>
> > **W3: Need case studies on model outputs for specific GDP and correponding SAT problems.**
>
> Thank you for your comment and suggestion. **We have added case studies in the revised [PDF](https://openreview.net/pdf?id=elmTU101oS) (in Appendix B.3).**
>
> For GDP problems, the model’s output is a binary value (0 or 1) at the instance level. For example, in the case of the k-clique problem, the input is a graph, and the output indicates whether the graph contains a clique of size k. If such a clique exists, the output is 1; otherwise, it is 0.
>
> Similarly, for the corresponding SAT problem (e.g., the k-clique problem transformed into a CNF formula), the output represents whether the formula is satisfiable. If the formula is satisfiable, the output is 1; otherwise, it is 0. The satisfiability result directly corresponds to the solution of the original GDP problem. For example, a satisfiable formula indicates that the original graph contains a clique of size k.
>
>
> We hope this response could help address your concerns and wish to receive your further feedback soon.

---

> > ### Comment · Reviewer_qz26 · 2024-11-22
> >
> > I appreciate the authors' efforts on explaining the significance of this work, as well as the additional experiments. I have no further comments.

---

> > > ### Author Response · Authors · 2024-11-22
> > >
> > > Dear Reviewer qz26,
> > >
> > > We extend our gratitude for your generous support and for helping us improve the paper. We appreciate your valuable suggestions.
> > >
> > > Best regards,
> > >
> > > The Authors

---

### Official Review · Reviewer_icFd · 2024-11-03

**Soundness:** 1
**Presentation:** 2
**Contribution:** 1
**Rating:** 3
**Confidence:** 5

**Summary:**

This paper proposes a novel paradigm CORAL that treats each CO problem type as a distinct modality and unifies them by transforming all instances into representations of the fundamental Boolean satisfiability (SAT) problem. The approach aims to capture the underlying commonalities across multiple problem types via cross-modal contrastive learning with supervision, thereby enhancing representation learning.

**Strengths:**

1.	This paper attempts to address an important issue, which is training a pre-trained model for the SAT domain using data from different fields. However, there is controversy over whether this problem is solvable, as relevant studies have shown that the transferability between different SAT problems is poor [1], and SAT problems are highly sensitive to structure; minor modifications to the structure of the SAT model can lead to changes in results [2]. Additionally, SAT formulas lack extensive node features, while GNN model predictions entirely depend on structure and features, which raises concerns about the ability of GNN models to serve as cross-field pre-trained models for SAT.
2.	The experimental results presented in this paper are quite good. However, these results are only compared with GCN and NeuroSAT models, neither of which are state-of-the-art (SOTA) models for learning GDP problem representations and SAT formula representations. This suggests that the baseline models used are too weak.
3.	The writing in this paper is excellent, with no noticeable grammatical errors.

[1] Li Z, Guo J, Si X. G4satbench: Benchmarking and advancing sat solving with graph neural networks[J]. arXiv preprint arXiv:2309.16941, 2023.

[2] Shi Z, Li M, Khan S, et al. Satformer: Transformers for SAT solving[J]. arXiv preprint arXiv:2209.00953, 2022.

**Weaknesses:**

1.	From the problem description, I believe that SAT is merely a format for problem representation rather than a modality; the k-clique and k-color instances used in the paper are just SAT problems derived from different domains, not different modalities. Moreover, there are existing toolkits such as CNFGen [1] that can convert problems like k-clique and k-color into SAT problems, so the second claimed contribution of this paper does not hold. .
2.	In terms of applicability, I do not think that all CO (Combinatorial Optimization) problems can be transformed into SAT problems; some complex CO problems with continuous variables need to be converted into Mixed Integer Programming (MIP) problems. Therefore, the universality of this method is quite limited.
3.	Regarding the optimization objective, the paper simply treats GDP problems and their corresponding SAT problems as positive examples, while other SAT problems within the same domain are treated as negative examples. This contrastive learning approach is overly simplistic and can easily generate false negatives because these SAT formulas may differ structurally but be logically equivalent or share the same satisfiability status.
4.	In terms of the model architecture, the paper addresses all GDP problems using a Graph Convolutional Network (GCN), which is a less precise method for solving combinatorial optimization problems [2][3], leading to low-quality learned representations. From an experimental standpoint, this implies that the baseline model is very weak. For SAT problems, the paper also employs the classic NeuroSAT model in a straightforward manner. Thus, I believe the rationality and innovation of the model are insufficient and do not effectively address the challenge of collaborative optimization across domains. Additionally, adopting different models for problems in each domain significantly increases the number of model parameters, affecting the scalability and generalization capability of the model to new domain problems (such as the Pigeonhole principle).

[1] Lauria M, Elffers J, Nordström J, et al. CNFgen: A generator of crafted benchmarks[C]//Theory and Applications of Satisfiability Testing–SAT 2017: 464-473.

[2] Georgiev D G, Numeroso D, Bacciu D, et al. Neural algorithmic reasoning for combinatorial optimisation[C]//Learning on Graphs Conference. PMLR, 2024: 28: 1-28: 15.

[3] Veličković P, Ying R, Padovano M, et al. Neural Execution of Graph Algorithms[C]//International Conference on Learning Representations 2020.

**Questions:**

1.	How do you justify the feasibility of constructing a cross-domain pre-training model for GDP problems based on GNN? Given that the distribution of GDP problems varies significantly and the prediction results heavily rely on logical reasoning, which is not a task that GNN excels at. For example, minor modifications to the structure of the SAT model can lead to changes in results. To answer my question, I suggest the authors to discuss these challenges more explicitly in the paper and propose ways to incorporate logical reasoning capabilities, or analyze the sensitivity of the model to structural changes.
2.	Why did you choose to use GCN (or GraphSAGE) as the baseline model for GDP representations? For GDP problems, these two models are not state-of-the-art; instead, they have been pointed out by existing work to be very weak baseline models. Since this paper does not introduce innovations at the model level but instead attempts to propose a universal architecture, you need to use SOTA graph learning models (stronger GNNs and graph transformers, such as PGN[1] and GraphGPS[2]) as backbones to validate the effectiveness of the architecture you have proposed.

[1] Cappart Q, Chételat D, Khalil E B, et al. Combinatorial optimization and reasoning with graph neural networks[J]. Journal of Machine Learning Research, 2023, 24(130): 1-61.

[2] Wang T, Payberah A H, Vlassov V. Graph Representation Learning with Graph Transformers in Neural Combinatorial Optimization[C]//2023 International Conference on Machine Learning and Applications (ICMLA). IEEE, 2023: 488-495.

3.	When you train specific models for data from each domain, how do you generalize this pre-training model to data from different domains, rather than just to data of varying difficulty levels within the same domain? How can you explain the necessity of building a model for each domain? This not only limits the model's generalization ability but also greatly increases its complexity.
4.	Why did you use other SAT problems from the same domain as negative examples? The selection of positive and negative examples in contrastive learning is a critical issue, and this paper also lists it as one of the core challenges. However, the solution provided in this paper is too blunt and can easily generate pseudo-negative examples, as these SAT formulas might differ in structure but be logically equivalent or have the same satisfiability status. Therefore, I believe you need to design a contrastive learning positive and negative sample selection method that can better balance structure, function, and domain knowledge. For instance, in terms of functionality, you could refer to papers like FGNN[3]. Such an innovation could significantly enhance the contribution of your paper.

[3] Wang Z, Bai C, He Z, et al. Functionality matters in netlist representation learning[C]//Proceedings of the 59th ACM/IEEE Design Automation Conference. 2022: 61-66.

---

> ### Author Response · Authors · 2024-11-19
>
> We would like to express our sincere gratitude for thoroughly evaluating our paper and providing insightful and constructive feedback. We are genuinely committed to addressing your concerns and respond to your specific comments below.
>
> > **W1.1: SAT is merely a format for problem representation rather than a modality.**
>
> Thank you for your valuable comment. We agree with your point that SAT can be seen as a representation format. However, we would like to clarify our use of the term "modality." In our context, we do not intend to imply that the problems are different modalities in the strict sense of the term, but rather that they represent different forms of a higher-level underlying difficulty. These instances—while rooted in SAT—can be viewed as different manifestations of a more general combinatorial optimization challenge, which is analogous to how multiple tasks in image and text domains are viewed as distinct modalities in current research. By framing these problems in this way, we hope to draw an analogy to the multi-modal models used in image and text, where the "modalities" are distinct but share a common underlying structure.
>
> **We have revised the [PDF](https://openreview.net/pdf?id=elmTU101oS) to further explain how we apply the term "modality" in this broader sense (in Sec. 3.2).**
>
>
> > **W1.2: The second claimed contribution of this paper does not hold.**
>
> Thank you for your comment. We would like to emphasize that the primary contribution of our paper is not merely about training a multi-distribution SAT model. Instead, our second key contribution focuses on using SAT as a bridge for information transfer across models from different problem domains.
>
> By using SAT as a common format, we aim to capture a higher-level, generalizable representation that can improve the performance of solving various CO problems, rather than just converting individual instances.
>
>
> > **W2: Not all CO problems can be transformed into SAT problems.**
>
> Thank you for your thoughtful comment. We fully acknowledge that the approach presented in our paper may not apply universally to all CO problem types. However, we would like to highlight that, of the 21 NP-complete problems identified by Karp (2010)[1], 10 are GDPs. These problems form the core of many challenges in the CO field.
>
> Also, as mentioned in the Conclusion section of our paper, exploring additional methods to handle other types of CO problems—such as those involving continuous variables—is part of our future research agenda. We plan to extend the applicability of our framework in subsequent studies.
>
> [1]Karp R M. Reducibility among combinatorial problems[M]. Springer Berlin Heidelberg, 2010.
>
> > **W3 & Q4: The contrastive learning approach can easily generate false negatives as SAT formulas may differ structurally but be logically equivalent or share the same satisfiability status.**
>
> Thank you for your valuable feedback. We would like to provide some clarification.
>
> 1. While SAT instances can share the same satisfiability status (either satisfiable or unsatisfiable), this does not imply that these instances will have similar representations. There are only two satisfiability statuses, but the number of possible SAT instances is infinite, leading to a wide variety of structural differences within the same satisfiability status.
> 2. The likelihood of encountering logically equivalent SAT instances in our setup is very low. We ensure that all original graphs in the same domain have distinct structures, and the SAT instances we generate have hundreds to thousands of clauses. Logically equivalent SAT formulas are very rare in such a scale unless they are specifically constructed. Moreover, any potential issues arising from logically equivalent SAT instances would only affect training when these instances appear in the same batch. The occurrence of such situations is extremely rare and would have minimal impact.
> 3. We present additional experimental results to show the impact of the issue. **We plot the contrastive loss curve (Figure 4)**, which demonstrates a smooth trajectory during training. **Additionally, we have introduced a new experiment that avoid such false negative samples through filtering negative samples by their satisfiability status (in Appendix D.6).** The results indicate that the impact from false negatives is minimal.

---

> > ### Comment · Reviewer_icFd · 2024-12-02
> >
> > Thanks for your reply. I still keep my score.

---

> > > ### Author Response · Authors · 2024-12-02
> > >
> > > Dear Reviewer icFd,
> > >
> > > Thank you for your response and for taking the time to review our rebuttal. While we respect your decision to maintain your original score, we would appreciate any specific feedback or suggestions that could help us improve the paper further. If you have any additional comments or areas of concern, we would be grateful for the opportunity to address them.
> > >
> > > Thank you again for your time and consideration.
> > >
> > > Best regards,
> > >
> > > The Authors

---

> ### Author Response · Authors · 2024-11-19
>
> > **W4.1 & Q2: The baseline model is very weak. The rationality and innovation of the model are insufficient.**
>
> Thank you for your thoughtful feedback. We would like to provide some clarifications to address your concerns.
>
> 1. For both the SAT and graph-based problems, we employed two different backbone models to demonstrate the robustness and effectiveness of our proposed framework. Importantly, the performance improvements we observe are consistent across these models, suggesting that the gains are not dependent on any specific model architecture.
> 2. For the SAT problems, we utilize the backbone model from G4SATBench[2], which has demonstrated high performance in SAT-based tasks. Our experimental results indicate that this model achieves close to 100% accuracy, which is a strong result, and the SAT model performance still benefits from our framework. Regarding the graph model, we adopt GCN and GraphSAGE because they are most widely used backbones in graph-based tasks. To address your concern, **we also conducted experiments with the two models you referenced[3][4], which are more precise for combinatorial optimization (in Appendix D.5).** The performance improvements we observed were consistent, further supporting the robustness of our framework, regardless of the specific model used.
> 3.  We would like to emphasize that the key innovation of our work lies not in the individual models themselves but in the cross-domain optimization enabled by using SAT as a unifying bridge. This novel approach allows the models to learn higher-level representations and underlying structures shared among different domains, which we believe represents a significant contribution to the field of combinatorial optimization.
>
>
> [2]Li Z, Guo J, Si X. G4satbench: Benchmarking and advancing sat solving with graph neural networks[J]. arXiv preprint arXiv:2309.16941, 2023.
>
> [3] Veličković P, Buesing L, Overlan M, et al. Pointer graph networks[J]. Advances in Neural Information Processing Systems, 2020, 33: 2232-2244.
>
> [4] Rampášek L, Galkin M, Dwivedi V P, et al. Recipe for a general, powerful, scalable graph transformer[J]. Advances in Neural Information Processing Systems, 2022, 35: 14501-14515.
>
>
> > **W4.2 & Q3: Adopting different models for each domain increases the number of model parameters, affects the scalability and generalization capability to new domain problems.**
>
> Thank you for your insightful comment.
>
> The reason we use separate models for each domain, particularly for GDPs, is that these problems often have strong dependencies on specific properties of the graphs. It is challenging to design a single, universal model that effectively captures all the necessary features across different problem domains. In order to create a model that can handle all types of problems, one would need to design domain-specific features for each problem, which requires significant expert knowledge and effort.
>
> To solve this challenge, our framework treats the graph problem instances as a mapping from graph to decision, allowing the neural network to automatically learn the relevant properties of the graphs needed to solve the problem. This approach eliminates the need for manually designing features for each specific problem. Thus, while each model is domain-specific, the framework unifies the training and the problem-solving process for all domains.
>
> We acknowledge that it is challenging to directly generalize the graph model to a completely new domain. However, **we have added new experiments to assess the generalization ability of our graph models on related tasks (in Appendix D.7.1).**
>
> Moreover, for cross-domain generalization, the SAT model has shown significant improvements. Specifically, when we apply our pre-trained SAT model to data from unseen domains (or even transfer it to other tasks), we observe that fine-tuning the pre-trained model on this new data leads to faster convergence and higher accuracy compared to starting from scratch with a SAT model trained directly on the new domain. These results are consistent with our framework’s ability to generalize well across domains, and they demonstrate the benefits of the cross-domain transfer capability.

---

> ### Author Response · Authors · 2024-11-19
>
> > **Q1: Justify the feasibility of constructing a cross-domain pre-training model for GDP problems based on GNN.**
>
> Thank you for your valuable comment and insightful suggestion. We would like to clarify this issue from the following points:
>
> 1. The SAT instances are represented as LCG/VCG graphs, where the logical reasoning involved in SAT problem-solving is explicitly encoded within the graph structure. This structure effectively captures the reasoning process needed to solve the SAT problem by converting the reasoning into feature learning on the graph.
> 2. The backbone we employed for the SAT model is specifically designed for SAT solving tasks and has demonstrated strong performance in related tasks. **We add a new experiment to show that the SAT model is sensitive to minor change of the graphs (in Appendix D.8).**
> 3. For GDPs, despite the limitations of GNN for capture such high-level features, our proposed contrastive learning framework, which align all graph models with the general, high-performance, and structure-sensitive SAT model, can effectively alleviate the limitations and enhance models' ability to handle the structural variations inherent in GDPs. **We also evaluate the sensitivity of our graph models to minor change of the graphs (in Appendix D.8).**
>
>
> We hope this response could help address your concerns and wish to receive your further feedback soon.

---

> ### Author Response · Authors · 2024-11-25
>
> Dear Reviewer icFd,
>
> We extend our sincere appreciation for your time and the valuable insights you've provided. We are eager to ensure that our rebuttal adequately addresses all your concerns and are looking forward to engaging in a further discussion with you. Should any lingering questions persist or if further clarifications are needed, please don't hesitate to reach out.
>
> Best regards,
>
> The Authors

---

> ### Comment · Area_Chair_nLqp · 2024-11-25
> **ICLR Public Discussion Phase Ending Soon**
>
> Dear Reviewer,
>
> This is a kind reminder that the discussion phase will be ending soon on November 26th. Please read the author responses and engage in a constructive discussion with the authors.
>
> Thank you for your time and cooperation.
>
> Best,
>
> Area Chair

---

> ### Author Response · Authors · 2024-11-27
>
> Dear Reviewer icFd,
>
> Thank you again for your valuable feedback on our submission. We appreciate the time and effort you’ve taken to evaluate our work.
>
> We understand that reviewing can be a time-consuming process, and we want to kindly follow up regarding the rebuttal we submitted a week ago. If you have any further questions or need clarification from us, we would be happy to provide additional information. However, if there are no further concerns, we would kindly request that you reconsider the rating in light of the clarifications we’ve provided.
>
> Thank you for your attention, and we look forward to hearing from you soon.
>
> Best regards,
>
> The Authors

---

### Author Response · Authors · 2024-11-20

Dear Area Chairs and Reviewers,

We extend our gratitude to all reviewers for their diligent efforts, insightful comments and constructive suggestions. Overall, the reviewers deem our work as 'pioneering' (qz26), 'novel' (icFd, qz26, 95cx, MBJq), 'effective' (qz26, 95cx, MBJq) and 'well-written' (icFd, qz26). We deeply appreciate their feedback and have made significant efforts during the rebuttal period to address their concerns and provide additional experimental results to strengthen our submission. The summary of our main efforts is presented as follows:

1. We conduct experiments using two additional GNN backbones that are more effective at learning graph instance representations. The experimental results show the consistent performance improvement on both backbones with our CORAL, and further prove its effectiveness. **(Appendix D.5)**
2. We implement an alternative contrastive loss function with specifically designed negative sampling strategy that eliminates false negative possibilities and include it into the comparison. The results indicate comparable performance to the original approach, suggesting that the impact of potential false negatives is minimal. **(Appendix D.6)**
3. We extended our evaluation on the graph models to include tasks more closely aligned with practical CO problems. The results show that the graph models generalize effectively not only across data of varying difficulty levels within the same domain but also across related tasks. **(Appendix D.7.1)**
4. We evaluate the generalization ability of the graph models on large-scale data, and show that CORAL can train scalable models effectively, highlighting its applicability to real-world, large-scale datasets. **(Appendix D.7.2)**
5. We generate subtle instances by minorly perturbing the structures of the original graphs to reverse the satisfiability status, and evaluate the performances of our models on these instances. The results reveal that CORAL significantly improves model performance on these challenging cases, demonstrating its ability to handle nuanced changes in graph structures. **(Appendix D.8)**
6. We conduct an additional ablation study on the cross-domain information transfer mechanism of CORAL. The experimental results highlight the critical role of cross-domain information in improving graph CO instance representation learning. **(Appendix D.9)**
7. We further clarify our use of the term 'modality' to prevent any misunderstanding. **(Section 3.2)**
8. We add case studies on model outputs for specific GDP and correponding SAT problems for a better illustration. **(Appendix B.3)**
9. We add an analysis of the computational cost associated with our experiments, further supporting the practicality of our approach. **(Appendix D.2)**

In our individual rebuttal, we provide thorough explanations and responses to each of the reviewers' questions. We aim for these responses to effectively address the concerns raised by the reviewers, and we are open to further discussion to ensure a thorough evaluation of our work.

---

### Meta-Review · Area_Chair_nLqp · 2024-12-19

**Metareview:**

This paper introduces a paradigm that treats each combinatorial optimization (CO) problem type as a distinct modality, unifying them by converting all instances into representations of the fundamental Boolean satisfiability problem. Experiments on seven graph decision problems demonstrate the method's effectiveness.

However, the reviewers have pointed out some important weaknesses. First, additional baselines are necessary for a comprehensive comparison. Second, not all CO problems can be transformed into SAT problems, limiting the method's universality. Therefore, I will not recommend accepting this paper in its current state.

**Additional Comments On Reviewer Discussion:**

Reviewers icFd, qz26, 95cx, MBJq rated this paper as 3: reject (keep the score), 6: marginally above the acceptance threshold (keep the score), 6: marginally above the acceptance threshold (raise to 6), and 3: reject, respectively.

The reviewers raised the following concerns.

- The model seems limited to solving a narrow class of GDP problems (Reviewers icFd and qz26).
- The author does not compare with sufficient baselines and stronger backbone GNNs (Reviewers MBJq, qz26 and 95cx).
- The feasibility of a GNN-based cross-domain pre-training model for SAT problems due to structural sensitivity (Reviewer icFd).
- The complexity of the framework and the lack of computational cost analysis (Reviewer MBJq).

By using additional experiments and providing more details in the rebuttal, the authors address some concerns about scalability, insufficient baselines, and complexity analysis. However, some fatal weaknesses have not been properly addressed by the authors' rebuttal. First, the applicability of this method is limited, as this method can only be applied to GDP problems that can transformed into SAT problems. Second, the paper does not include sufficient baselines in their experiments.



Therefore, I will not recommend accepting this paper in its current state.

---

### Decision · Program_Chairs · 2025-01-22

Reject